# Augmenting the German weather radar network with vertically pointing cloud radars: implications of resolution and attenuation

Christian Heske [1], Florian Ewald [1], and Silke Groß [1]

[1]Deutsches Zentrum für Luft- und Raumfahrt (DLR), Institut für Physik der Atmosphäre, Oberpfaffenhofen, Germany

**Correspondence:** Christian Heske (christian.heske@dlr.de)

**Abstract.** A beam-aware columnar vertical profile (BA-CVP) methodology that incorporates data of the national German operational radar network with the aim of augmenting vertically pointing cloud radars is introduced. The method uses polarimetric radar data collected in plan position indicator (PPI) scans of multiple operational radars and considers the contributions of each operational radar in a beam-aware manner. The results of the method are paired with measurements of a vertically pointing cloud radar located in range of the operational radars and compared to measurements of dedicated scanning radars collected in range height indicator (RHI) scans for two different case studies. The combination of side-looking operational radars in the BA-CVP format with a vertically pointing cloud radar allows the simultaneous exploitation of polarimetric multi-frequency observations including radar variables like the differential reflectivity, the dual-wavelength ratio, Doppler fall speed and linear depolarisation ratio measurements to study the microphysical properties of hydrometeors while not having to rely on the availability of dedicated scanning radars. The extracted BA-CVPs based on operational data deliver good results with respect to resolving finer details in radar reflectivity and differential reflectivity when compared to measurements of dedicated radars. Usage of differential phase measurements is discussed as marker for high hydrometeor attenuation affecting the measurements of radars far from the point of interest. For future use of this method, the coverage of the national German operational radar network is studied and recommendations for locations with potential for additional vertically pointing cloud radars are pointed out.

## 1 Introduction

The application of radars has proven to be essential for comprehensive studies of clouds and precipitation (Trömel et al., 2021). Enhancing the understanding of microphysical processes and properties deep within clouds, polarimetric radar measurements provide invaluable information about hydrometeor characteristics which no other remote sensing technique can reach (Kumjian, 2013). This can be of exceptional use for improving the representation of ice particles in existing microphysical schemes in numerical models which still turns out to be a major uncertainty in the modeling of weather and climate (Morrison and Milbrandt, 2015; Xue et al., 2017; Morrison et al., 2020).

Polarimetric radar variables like the differential reflectivity $Z_{dr}$ or the differential phase $\Phi_{DP}$ are sensitive to the non-sphericity of hydrometeors that are ideally captured from an oblique perspective at low radar elevation angles since they are mostly aligned horizontally. To aid the understanding of microphysical processes and to further constrain the properties of hydrome-

teors, polarimetric measurements are often combined with multi-frequency and Doppler fall-speed observations preferentially taken at close range by vertically pointing remote sensing equipment like cloud radars. To tackle this inherent observational dilemma which requires two or more radar systems in different measurement geometries to facilitate the simultaneous exploitation of polarimetry, multi-frequency and fall-speed measurements (Kneifel et al., 2015, 2016; Oue et al., 2018, 2021), different

approaches concerning the measurement strategy and data processing chain have been developed, depending on site and dedicated equipment availability. In the case of two or more dedicated radars at spatially separated sites, coordinated range height indicator scans (RHI) from one or all radars towards each other are possible and can either be compared directly to each other (Tetoni et al., 2022) or vertical profiles can be extracted from the RHIs of the scanning radars (Andrić et al., 2013; Moisseev et al., 2015; Tiira and Moisseev, 2020; Gehring et al., 2020; Blanke et al., 2023). In most cases however all instruments are

concentrated at one measurement site increasing the difficulty to obtain polarimetric observations from a side-viewing perspective that can be complemented with the vertical profiles provided by zenith pointing cloud radars.

Initially introduced as a way of decreasing noise and therefore increasing the signal-to-noise-ratio of radar variables measured by plan position indicator scans (PPI) of mostly operationally used polarimetric weather radars with fast azimuthal scan speeds, so-called quasi-vertical profiles (QVP) have proven as a method to partly overcome these difficulties. To create a QVP

the azimuthal median (Kumjian et al., 2013) or azimuthal average (Ryzhkov et al., 2016) of a PPI scan conducted at a specific elevation angle is calculated for every range gate and then projected to a regular vertical grid forming a vertical profile. This procedure can be repeated for each consecutive PPI scan at the same elevation angle creating data in a time-height-format which can then be easily paired with time-height data of vertically pointing cloud radars. Since the scanning radar is not pointing vertically during the PPI scans, polarimetric variables are collected from an oblique perspective. Because of the conical

shape of each PPI scan used for computation, the diameter of the area covered by QVPs increases from several kilometers at low altitudes to several tens of kilometers at higher altitudes depending on the elevation angle of the PPI scan. Using the measurements of a PPI scan at an elevation angle of $20°$ for example leads to a cone diameter of already 16.5 km at 3 km altitude which increases to 55 km at 10 km altitude (Ryzhkov et al., 2016). Thus, QVPs are mainly useful for horizontally homogeneous weather situations like large-scale stratiform precipitation at or close to the location of the PPI-performing radar.

QVPs have so far been used in a variety of different situations like hydrometeor classification (Lukach et al., 2021), study of microphysical properties of snow (Griffin et al., 2018; Oue et al., 2021) and melting layer detection (Trömel et al., 2019; Griffin et al., 2020).

Operational weather radars like the S-band WSR-88Ds of the American NEXRAD network or the C-band DWSR5001C/SDP-/CEs operated by the Deutsche Wetterdienst (German Meteorological Service, DWD) in Germany often follow a scan strategy

consisting of multiple PPI scans performed at different elevation angles within quick succession of each other. This PPI scan sequence is then repeated in regular intervals. The method of QVPs can be extended to utilize all PPI scans within one scan sequence of operational or dedicated radars forming so-called range-defined QVPs (RD-QVP) (Tobin and Kumjian, 2017). Here, traditional QVPs are computed for all available PPIs of one scan sequence for data points within a pre-defined range from the radar which are then all used during interpolation to a regular vertical grid with inverse distance weighting. Preserving

the reduction in statistical uncertainty from traditional QVPs by azimuthal averaging, range-defined QVPs have increased ver-

tical resolution by making use of all available PPI scans in one scan sequence. By setting a range of interest around the radar and weighting the data points by inverse distance, the influence of the more remote areas is reduced and the resulting vertical profiles consist of values more close to the desired location. Just like with regular QVPs, the extracted profiles of RD-QVPs remain centered at the location of the PPI scanning radar. This combined with the fact that both QVP methods lose spatial

information due to the 360° azimuthal averaging, limit the use of the QVP methods in combination with vertically pointing measurement equipment further away from the PPI scanning radar since the extracted QVPs might not adequately represent the measurements of the off-site equipment. RD-QVPs have been used for similar topics as the original QVPs like precipitation estimation (Bukovčić et al., 2020; Ryzhkov et al., 2022) and melting layer detection (Ryzhkov and Krause, 2022).

In many cases only a vertically pointing cloud radar is available at a specific measurement site and no additional dedicated

radar at another frequency for RHI scans towards the cloud radar is available or in range. Nonetheless these single-radar sites are often in range of one or several operationally used weather radars that perform PPI scan sequences in regular intervals as mentioned previously. Methods like GridRad (Homeyer and Bowman, 2017) that combine the data of multiple operational radars and map it to a regular cartesian grid which could then readily be used to extract vertical profiles at the location of the cloud radar usually only produce composites with a limited horizontal and vertical resolution unable to depict finer details

(Murphy et al., 2020). Additionally no azimuthal averaging or other noise-reducing procedure except usage of multiple radars that have overlapping measurement ranges is done which further reduces the usability of the data for combination with a vertically pointing cloud radar of high resolution. The aforementioned QVPs or range-defined QVPs as radar-centric methods are also not suitable to provide additional data since the distance between the radars providing QVPs and the cloud radar might reach several tens of kilometers creating a considerable spatial mismatch between cloud radar data and the processed QVPs. To

overcome these limitations, slanted vertical profiles (SVPs) and enhanced vertical profiles (EVPs) were introduced (Bukovčić et al., 2017). The creation of SVPs and EVPs is based on a segment chosen in azimuth and range centered at the location of interest that is cut out of the volume data provided by PPI scans. This segment is kept small as SVPs and EVPs were initially proposed for comparison to disdrometer data. Similar to the QPV methods, SVPs and EVPs both include azimuthal averaging for noise reduction within the specified segment. SVPs rely on a single elevation angle and therefore lead to only very shallow

profile data. In contrast EVPs take all available elevation angles into account and calculate the median for a limited number of range positions around the location of interest. Since SVPs rely on only one elevation, random noise is still a considerable factor if operational weather radar data is used. Additionally both methods depend on small local segments further decreasing statistical accuracy.

Extending EVPs spatially, Murphy et al. (2020) then proposed the method of extracting columnar vertical profiles (CVPs)

which sources data from a much larger segment. After azimuthal averaging and favoring data points more closely to the desired location, an inverse distance weighted moving averaging technique is applied which maps the data to a regular vertical grid. This leads to a considerable improvement in the reduction of noise as well as an increase in vertical resolution. The method however can also lead to the occurrence of gaps in the extracted profiles where the vertical distance between data points of successive elevation scans is bigger than the moving average bin size (Murphy et al., 2020; Bukovčić et al., 2020).

This severely worsens the quality of the extracted profiles at long distances away from the PPI scanning radar or when only

a limited number of elevation scans is available. CVPs are often complemented by QVPs or RD-QVPS and used to study different parts of fast moving atmospheric phenomena like hurricanes (Hu et al., 2020; Hu and Ryzhkov, 2022) or to retrieve microphysical properties of ice hydrometeors via polarimetry, Doppler fall-speed and dual-wavelength ratio observations (Oue et al., 2025).

To overcome the limitation of the CVP method where gaps occur at long ranges when the vertical distance between data points of successive elevation scans gets too large, we propose enhancements to the CVP methodology of Murphy et al. (2020) where after azimuthal averaging of the chosen segment, the physical extent of each data point based on range gate length and beamwidth is taken into account. By introducing correct handling of beam broadening, the extraction of so-called beam-aware CVPs (BA-CVPs) with no artificial gaps regardless of moving average bin size and distance from the PPI scanning radar is 105 ensured.

The procedure can be extended to any other radar system capable of vertical pointing within the range of one or several operational radars. Figure 1 for example shows the locations of all 17 operational radars that are part of the German DWD network. It also depicts the position of Cloudnet sites belonging to the ACTRIS research infrastructure (Laj et al., 2024) with active cloud radars which provide vertical profiles on a regular basis and can also feature other vertically pointing measurement equipment 110 like ceilometers or Lidar systems. The high concentration of vertically pointing cloud radars and the dense network of operational C-band radars in south Germany together with a number of dedicated scanning radars facilitate a unique opportunity to study the possibility of augmenting vertically pointing cloud radars with data of operational weather radars.

To investigate the usability of BA-CVPs based on PPI data, BA-CVPs extracted from data of two operational radars are compared with vertical profiles from RHIs performed by dedicated scanning radars for two different case studies. Additionally 115 all vertical profiles are compared against high-resolution data provided by a vertically pointing cloud radar. The present paper aims to advance the following scientific objectives:

- Combining polarimetric multi-frequency and Doppler fall-speed observations to enhance the understanding of micro-physical properties of cloud hydrometeors.

- Overcoming existing challenges when different radar equipment is combined with regard to equipment availability and 120 measured radar variables.

- Development of an approach based on operational radar data that incorporates adequate consideration of beam-broadening and uses data of multiple operational radars in range of a point of interest.

- Exploitation of Germany's dense operational polarimetric radar network and high concentration of vertically pointing radars together with a number of dedicated scanning radars as a testing ground for enhanced data processing methods 125 like BA-CVPs that aim to complement vertically poiting cloud radars with operational weather radar data.

The paper is structured as follows: Section 2 introduces all involved radar systems with their technical specifications. In section 3 the methods for extracting vertical profiles from RHI scans of dedicated scanning radars and BA-CVPs from PPI scans done by operational radars belonging to the national German radar network are explained in detail. Section 4 contains the analysis of

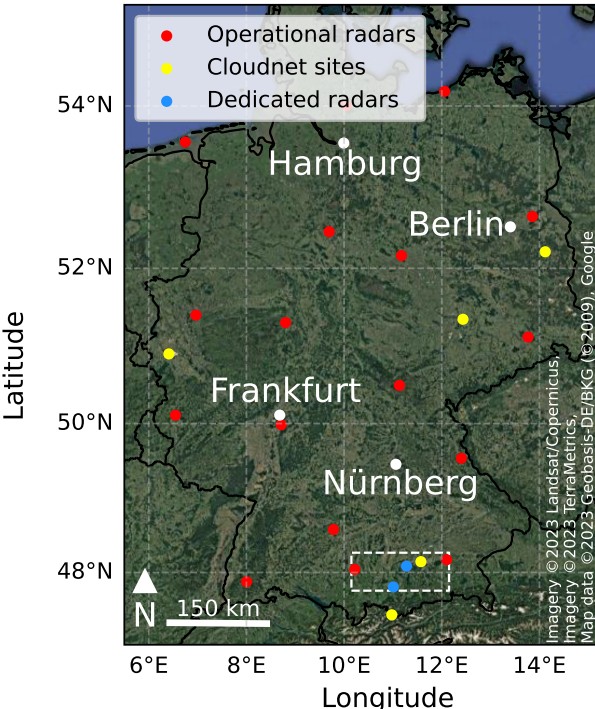

**Figure 1.** Map of Germany which shows the location (red) of all 17 radars in operational use within the German weather radar network managed by the German Meteorological Service. Shown are also all operational cloudnet sites (yellow) and the two dedicated radar systems (blue) utilized for comparison. The area marked by the white rectangle contains all radars used in this study. Zoom-ins for the latter are found in Fig. 3 and 4, respectively. Imagery ©2023 Landsat/Copernicus, Imagery ©2023 TerraMetrics, Map data ©2023 Geobasis-DE/BKG (©2009), Google.

extracted BA-CVPs from operational data, extracted vertical profiles from dedicated RHI scans and the profiles of a vertically pointing cloud radar for two different case studies. Different radar variables are presented and compared within the different extraction methods and datasets. The discussion is presented in section 5 and, finally, conclusions and final remarks for the presented approach are drawn in section 6.

## 2 Radars

In this study the synergistic use of spatially separated radar systems operating at different wavelengths and operation modes for two different case studies is discussed. In particular, the role of operational radar systems and the use of their data in combination with or as a substitute for data measured by dedicated radars is investigated when paired with the data of a vertically pointing radar operating at a different wavelength. To address this, the set of used radars consists of a cloud radar in the Ka-Band (MIRA-35), two dedicated C-band radars (POLDIRAD, MHP) performing oriented RHI scans towards the cloud

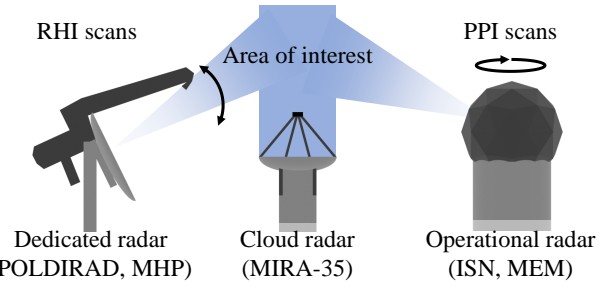

**Figure 2.** Schematic view of the measurement setup. The cloud radar MIRA-35 looks vertically while one of the dedicated radars (POLDIRAD, MHP) performs RHIs pointing towards MIRA-35 and the operational radars (ISN, MEM) operate within their usual scan strategy.

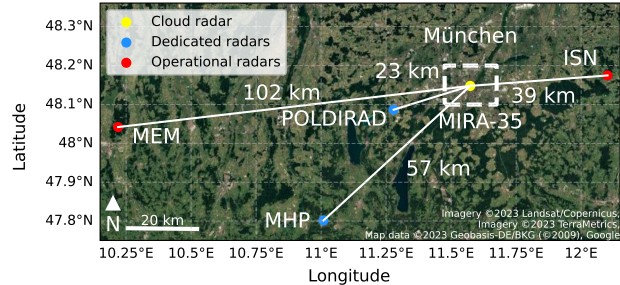

**Figure 3.** Map showing all radar locations relevant for this study. The depicted area is a zoom-in of the white rectangle in Fig. 1. The position of MIRA-35 (yellow dot) is in the center of Munich. Dedicated and operational radars are marked blue and red, respectively. The white rectangle marks the area of interest for this study. Imagery ©2023 Landsat/Copernicus, Imagery ©2023 TerraMetrics, Map data ©2023 Geobasis-DE/BKG (©2009), Google.

radar and two operational C-band radars (ISN, MEM) of the DWD radar network operating in their usual 5 minute scan strategy
(Seltmann et al., 2013). A schematic view of the measurement configuration with the different radars and their operation modes is shown in Fig. 2. The geographic region for this study can be seen in Fig. 1 marked by the white rectangle. A zoom-in of this region is shown in Fig. 3 (yellow: cloud radars, blue: dedicated radars, red: operational radars).

## 2.1 Vertically pointing cloud radar MIRA-35 at LMU

MIRA-35 is a scanning Doppler cloud radar developed by Metek (Meteorologische Messtechnik GmbH, Elmshorn, Germany)
operating in the Ka-band at a frequency of 35.2 GHz (8.5 mm) with a dish size of 1 m and a 3 dB beam width of 0.6°
(Görsdorf et al., 2015). It is located in the center of Munich on the roof of the Ludwig Maximilians University (LMU) at coordinates 11.57° lon, 48.18° lat at an altitude of 541 m a.m.s.l. and is operated by the Meteorological Institute Munich (MIM). The geographical location in relation to the C-band radars is shown in Fig. 3. In normal operation the magnetron of

MIRA-35 provides a peak power of 30 kW and transmits pulses with a pulse duration of 0.2 µs at a pulse repetition frequency (PRF) of 5000 Hz. This corresponds to a maximum range of 30 km and a range resolution of 30 m. MIRA-35 only emits horizontally polarized radiation but measures both vertical and horizontal components of the backscattered signal. Compared to the polarimetric C-band radars, MIRA-35 therefore is not able to measure radar variables that require the emission of horizontal and vertical polarized radiation like the differential reflectivity but still provides linear depolarisation ratio measurements. While MIRA-35 is capable of following complex scan patterns including RHI or PPI scans, the data used in this study was solely collected with the radar pointing vertically generating profiles with an average time of 10 s. The technical characteristics of MIRA-35 in comparison to the C-band radars are presented in Table 1.

## 2.2 Dedicated C-band radar POLDIRAD

POLDIRAD is a polarization diversity Doppler weather radar operating in the C-band at a frequency of 5.5 GHz (54.5 mm) with an offset parabolic antenna spanning 4.5 m and a circular beam width of 1° (Schroth et al., 1988). It is located on the roof of the Institute for Atmospheric Physics of the German Aerospace Center in Oberpfaffenhofen at coordinates 11.28° lon, 48.09° lat at an altitude of 603 m a.m.s.l. 23 km southwest of MIRA-35. The geographical location in relation to MIRA-35 and the other C-band radars is shown in Fig. 3. A particular feature of POLDIRAD is its advanced polarization network which allows the selection of transmitted and received polarization for every individual pulse. With it the transmission and reception of co- and crosspolar components of any incoming signal regardless of polarization type is possible (Reimann and Hagen, 2016). As dedicated research radar without operational obligations, the transmit mode characteristics of POLDIRAD can be varied freely within design limits. For this study POLDIRAD emitted pulses with a pulse duration of 1 µs at a pulse repetition frequency of 1150 Hz which amounts to a maximum range of 130 km and a range resolution of 150 m. The technical characteristics of POLDIRAD in comparison to MIRA-35 and the other C-band radars are presented in Table 1. A validation of the differential reflectivity calibration is given in Tetoni et al. (2022).

## 2.3 Dedicated C-band radar MHP

The research radar at the Meteorological Observatory Hohenpeißenberg (MHP) of the German Meteorological Service (DWD) is a dedicated research C-band Doppler radar with polarization diversity operating at a frequency of 5.64 GHz (53.15 mm) with a parabolic antenna spanning 4.27 m and a beamwidth of 1°. It is of the same type (EEC DWSR5001C/SDP/CE) as the 17 operational radars managed by the DWD and can be used operationally as substitute if one of the operational radars in the vicinity of Hohenpeißenberg is down for maintenance or repair. The main purpose of MHP however, is to test and evaluate new technologies like new radar data processing algorithms, radar software updates or new radar products before introduction into operational service (Frech et al., 2017). The geographical coordinates of MHP are 11.01° lon, 47.80° lat at an altitude of 1006.2 m a.m.s.l. with a distance of roughly 57 km southwest of Mira-35. Figure 3 shows the location in relation to the other radars. During scans MHP emits pulses with a pulse duration of 0.4 µs at a PRF of 2400 Hz leading to a maximum range of 61.5 km and a range resolution of 60 m. The technical characteristics of MHP in comparison to Mira-35 and the other C-band radars are shown in Table 1. Details about differential reflectivity calibration are given in Frech and Hubbert (2020).

**Table 1.** Technical details of MIRA-35, POLDIRAD, MHP, ISN and MEM in comparison.

| Parameter | MIRA-35 | POLIDRAD | MHP | ISN, MEM |
|---|---|---|---|---|
| Frequency (GHz ) | 35.2 | 5.5 | 5.640 | 5.625, 5.630 |
| Peak transmitted power (kW ) | 30 | 400 | 500 | 500 |
| Antenna diameter (m) | 1 | 4.5 | 4.27 | 4.27 |
| Beam width (°) | 0.6 | 1.0 | 1.0 | 1.0 |
| Pulse Duration (μs) | 0.2 | 1 | 0.4 | 0.4 or 0.8 |
| Pulse repetition frequency (Hz) | 5000 | 1150 | 2400 | 600 - 2410 |
| Max. range (km) | 30 | 130 | 61.5 | 60 - 180 |
| Range resolution (m) | 30 | 150 | 60 | 250 or 1000 |

## 2.4 Operational radars ISN and MEM

The German weather radar network managed by the DWD consists of 17 polarization diversity Doppler radars of type EEC DWSR5001C/SDP/CE operating in the C-band at frequencies between 5.6 and 5.65 GHz (53.06 - 53.53 mm) with parabolic antennas spanning 4.27 m and beamwidths of 1°. The locations of all network radars are presented in Fig. 1. With its coverage the DWD radar network is the foundation for monitoring the weather situation over Germany providing invaluable real-time data on precipitation and meteorological events. For this purpose the DWD network follows a five minute scan strategy consisting of a terrain-following PPI precipitation scan followed by ten PPI scans which sample the 3D volume around each radar at ten different elevation angles and finally a birdbath scan to calibrate the differential reflectivity (Frech et al., 2017; Frech and Hubbert, 2020). The data used in this study exclusively consists of the volume data collected during the 10 PPI scans. During these PPI scans and depending on the elevation angle of each sweep the radars operate with different azimuth turn speeds between 12 and 30 °s$^{-1}$, staggered PRFs between 600 and 2410 Hz and pulse durations of 0.4 or 0.8 μs. This amounts to a maximum range between 60 and 180 km but always with a set range resolution of 1000 m for the 28th of May 2019 and 250 m for the 8th of July 2021. A comprehensive overview of the radar parameters used for each individual PPI sweep is listed in Appendix A.

For the present study the data of two DWD network radars in the vicinity of MIRA-35 was used: The first radar (ISN) is located in Isen with coordinates 12.10° lon, 48.17° lat at 677.77 m a.m.s.l. east of Mira-35 with a distance of roughly 39 km. The location of the second radar (MEM) is Memmingen at coordinates 10.22° lon, 48.04° lat with 725 m a.m.s.l. roughly 102 km west of Mira-35. Figure 3 shows the location and Table 1 the technical details of the two radars in comparison to the other radars. Details about differential reflectivity calibration are given in Frech and Hubbert (2020).

## 3 Methods

### 3.1 Profile extraction

As stated previously, both, PPI and RHI scans, provide polarimetric radar data from an oblique perspective. The key differences between the two measurement strategies lie in the achievable vertical resolution and signal-to-noise-ratio, both of which are usually lower for the PPI scans because of the fast azimuthal scan speeds and limited number of measured elevation angles when compared to RHI scans. To examine the usability of PPI data for combination with a vertically pointing cloud radar, vertical profiles have to be extracted and compared to vertical profiles extracted out of RHI scans.

#### 3.1.1 Virtual profiles from RHI scans

For RHI scans the vertical profile extraction is a rather straightforward task as RHI scans are usually projected from the native spherical coordinates to cartesian coordinates for the purpose of being presented in a range-height plot. It is therefore only necessary to correctly calculate the distance between the C-band and Ka-band radar and then extract an interpolated vertical profile at the exact position of the cloud radar from the projected range-height plot. To improve the signal-to-noise-ratio, a radial range around the location of the Ka-band radar from the perspective of the C-band radar can be defined. All data points within the defined range are projected along the horizontal. To better compare the measurements of different radars and measurement types it is helpful to map the data to common and evenly spaced height values. For this purpose a height grid used for all C-band radars with a resolution of 50 m is defined starting at the geographic height of MIRA-35 going up to a chosen maximum height value of 11 km. On this grid a moving average with a bin size of 100 m for the dedicated radars is applied keeping measured features sharp in detail while also allowing a limited amount of smoothing and noise reduction. Instead of a single interpolated vertical profile at the position of MIRA-35, an averaged vertical profile with higher statistical accuracy can therefore be extracted. The differential reflectivity is calculated at the very end from extracted profiles of the horizontal and vertical reflectivity. It is well known (e.g. Ryzhkov et al. (2016); Ryzhkov and Zrnic (2019)) that instrumental noise affects $Z_{dr}$ and $\Phi_{dp}$ more significantly and is noticeable for even moderate signal-to-noise ratio. As all averaging is done in linear space, the impact of noise-related biases can be significant. To prevent measurements with low reflectivity and therefore low signal-to-noise ratio to influence the bin averages too strongly, $Z_{dr}$ and $\Phi_{dp}$ are weighted by the reflectivity during averaging. The bin average $\bar{X}_{bin}(h)$ at height $h$ containing $n(h)$ measurement values $x_i$ is therefore calculated as

$$\bar{X}_{bin}(h) = \sum_{i=1}^{n(h)} w_{z,i} x_i \tag{1}$$

where $w_{z,i}$ is a normalized weighting factor defined as

$$w_{z,i} = \frac{x_{Z,i}}{\sum_{j=1}^{n(h)} x_{Z,j}} \tag{2}$$

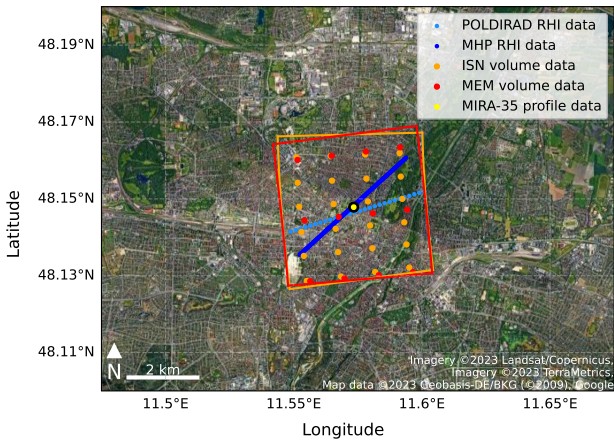

**Figure 4.** Map of the area of interest for this study. The image shows the metropolitan region of Munich as a zoom-in of the white rectangle in Fig. 3. The yellow dot marks the position of MIRA-35. The image also includes the measurement points of the dedicated (blue colors) and operational radars (red and orange) within their respective segment of roughly 4 km x 4 km centered on MIRA-35 at the lowest elevation angle. The data of POLDIRAD is not directly above MIRA-35 with a very minor offset which is the result of the pointing error associated with the pointing accuracy of POLDIRAD. Imagery ©2023 Landsat/Copernicus, Imagery ©2023 TerraMetrics, Map data ©2023 Geobasis-DE/BKG (©2009), Google.

with $x_{Z,i}$ being the reflectivity value of data point $x_i$.

All data points of the lowest elevation angle measured by POLDIRAD and MHP contributing to the averaged vertical profile are shown in Fig. 4 in light blue and dark blue, respectively. For the present study a distance of 2.1 km up- and downradial from the position of MIRA-35 for a total of 4.2 km segment length was chosen to match the size of the segment used for the extraction of BA-CVPs from PPI scans which is described in the next section.

### 3.1.2 Virtual profiles from PPI scans

A similar procedure like for the RHI scans can also be applied to the 3D volume PPI scan data of the operational DWD radars. From the original data in native spherical coordinates a virtual RHI scan at the azimuth angle where the operational radar is pointing towards the cloud radar is interpolated as often the exact location of the cloud radar lies between two measured azimuth angles. From this interpolated RHI scan the vertical profile can then be extracted as described before. It is however important to note that because of the fast azimuth turn speeds and limited pulse repetition frequency at more distant ranges, the

radar variables of operational radars often suffer from high noise especially for the polarimetric variables. Extracting a vertical profile from a single interpolated RHI scan, even when using the previously described method of defining a radial range for averaging, is therefore not always optimal or enough to achieve adequate statistical accuracy (Ryzhkov et al., 2016).

**Beam-aware CVP**

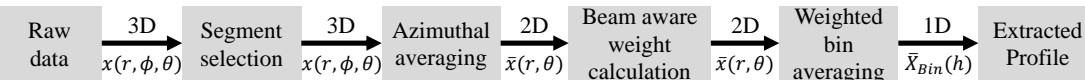

**Figure 5.** Schematic flow diagram with all major steps to extract BA-CVPs from raw data. The text above the arrows indicates the dimension of the data matrix written below the arrow that is passed from one processing step to the next.


BA-CVPs were developed for this study to allow time-height comparison between the data collected by operational radars in the form of PPI scans and the data of a vertically pointing cloud radar. Figure 5 depicts a flow chart containing all major processing steps necessary to compute a BA-CVP based on operational radar data. As the data of operational radars for each time step is three-dimensional volume data that will eventually be reduced to a one-dimensional profile, Fig. 5 also explains the

dimensions of the data matrix that is passed from one processing step to the next. A BA-CVP consists of a three dimensional segment defined in range, azimuth and height. For the purpose of this work the radial depth of the segment was chosen to be 4.2 km and centered at the location of MIRA-35 spanning 2.1 km upradial and 2.1 km downradial from the cloud radar. The azimuthal range of the segment was determined for ISN and MEM separately to achieve a segment that approximately resembles a square of similar size ( 16 km$^2$) for both operational radars. This was done to always ensure sampling of roughly the

same volume around MIRA-35 to establish a good basis for comparison with MIRA-35 and the dedicated C-band radars. The radial range of 4.2 km and therefore the segment area size of 16 km$^2$ used for this study proved to be a good balance between preventing the influence of horizontal heterogeneity and increasing signal-to-noise-ratios for the chosen case studies consisting of mostly stratiform precipitation. As a consequence of the square segment and the different distances between ISN/MEM to MIRA-35, the number of measurement points within the extracted segments differ for ISN and MEM. This can be seen in Fig.

4 where the segment outlines as well as all the data points contained in the chosen segments at the lowest elevation angle are plotted for ISN (orange) and MEM (red) in comparison to the RHI data points at lowest elevation. For better visibility data of 28th of May 2019 was chosen where the range resolution of the operational radars was still 1000 m instead of 250 m. It is evident that the ratio between measured azimuth angles contained in the segment for the operational radars is 1:2 which roughly resembles the ratio between the distances of ISN-MIRA-35 to MEM-MIRA-35 of 39 km : 102 km. Since the radial

resolution changes from 1000 m for the 28th of May 2019 to 250 m for the 08th of July 2021, the segment for the 08th of July 2021 includes roughly four times more data points. The effects of this will be discussed in section 5.

All measured data points within the defined segments are first averaged azimuthally bringing the data into RHI format. During this first averaging process all radar variables except the radar reflectivity itself are weighted by the radar reflectivity. This and all following averaging steps are done in linear space for each variable. As the operational radars provide volume data in

spherical coordinates, each raw data point $x(r,\phi,\theta)$ is uniquely identifiable by range $r$, azimuth angle $\phi$ and elevation angle $\theta$. Each azimuthally averaged data point $\bar{x}(r,\theta)$ can therefore be calculated by

$$\bar{x}(r,\theta) = \sum_{k=1}^{m} w_Z(r,\phi_k,\theta) x(r,\phi_k,\theta) \tag{3}$$

where $m$ signifies the total number of contributing data points in the azimuthal direction within the selected segment for each $r$ and $\theta$ and $w_{z,k}$ is a normalized weighting factor based on the measured horizontal reflectivity that is applied for all radar variables except the reflectivity itself. $w_{z,k}$ is defined by

$$w_Z(r,\phi_k,\theta) = \frac{x_Z(r,\phi_k,\theta)}{\sum_{l=1}^{m} x_Z(r,\phi_l,\theta)} \tag{4}$$

where $x_Z(r,\theta,\phi)$ is the measured horizontal reflectivity for each $r$, $\phi$ and $\theta$. To keep information about the statistical weight of each data point available, the number of data points contributing in the azimuthal averaging per elevation angle and range is tracked. Analogous to the handling of the projected RHI scans previously it is beneficial for adequate comparison to map the data to evenly spaced height values by interpolation or similar methods after transformation into cartesian coordinates. Because of the much lower data point density due to the limited amount of measured elevation angles during the PPI scans, a moving average technique with a small height bin resolution would leave a number of height bins without data points and therefore artificially add gaps in the extracted profile (Murphy et al., 2020; Bukovčić et al., 2020). This is especially true for segments that lie at great distance from the PPI performing radar system as the vertical distance between data points of neighboring elevation angles increases with range. These gaps between PPI scans are, however, mostly a display artifact. When the beam broadening is taken into consideration, the lower PPI scans are even slightly overlapping for the operational scan strategy. A simple increase in the height bin size of the moving average would only lead to more smoothing and therefore further decrease the already moderate resolution of PPI data. In contrast, BA-CVPs consider the physical extent of each data point based on range gate length and beamwidth as schematically indicated in Fig. 6. All data points are drawn centered within their respective range gate to model the physical extent. Range gates that are intersected by the height bin are colored in light orange for the lower elevation angle and light blue for the higher elevation angle. The areas of intersection between the height bin and each range gate are colored in dark orange for the lower elevation angle and dark blue for the higher elevation angle. To calculate the average value of the height bin, the ratio between the size of each individual range gate to the area of intersection belonging to the range gate is calculated and used as weight for the data point during the averaging. The average of a height bin at height $h$ containing $n(h)$ azimuthally averaged data points $\bar{x}(r,\theta) = \bar{x}_i$ is defined as

$$\bar{X}_{\text{bin}}(h) = \sum_{i=1}^{n(h)} w_i \bar{x}_i \tag{5}$$

where $w_i$ are the normalized weighting factors for each azimuthally averaged data point which can be written as

$$w_i = \frac{w_{b,i} w_{a,i} w_{z,i}}{\sum_{j=1}^{n(h)} w_{b,j} w_{a,j} w_{z,j}} \tag{6}$$

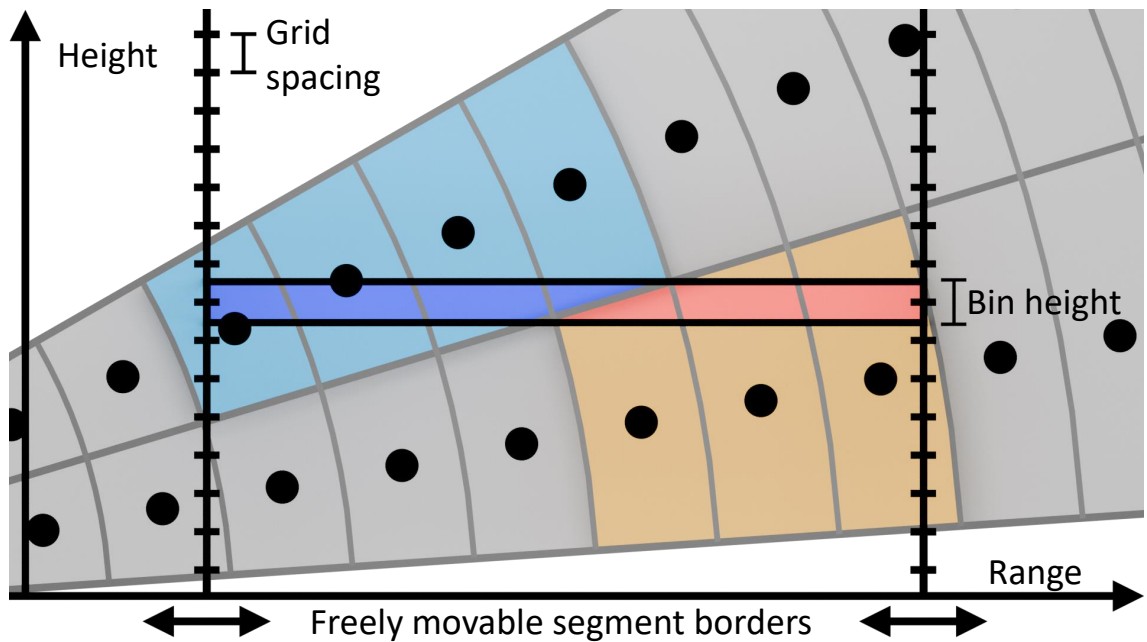

**Figure 6.** Schematic drawing showing the extraction of vertical profiles following the BA-CVP approach. Earth curvature and beam refraction are considered in the actual computations but neglected in the graphic for easier readability. Similarly only two out of the overall ten elevation angles are depicted. For each range gate intersected by the height bin (light blue, light orange) the size of the intersection with the height bin (dark blue, dark orange) is computed and used as weight during the moving average procedure.

consisting of 3 different, normalized weighting factors. The weighting factor $w_{b,i}$ is derived for each $\bar{x}_i$ from the area of

intersection between individual range gate and height bin as described previously. The second term $w_{a,i}$ is the weighting factor based on the azimuthal averaging step and is defined for each $\bar{x}_i$ as the number of contributing data points in the azimuthal averaging divided by the total number of contributing data points within the segment and height bin. This weighting factor is introduced to favor azimuthally averaged data points with a higher number of contributing raw data points to further reduce the influence of noise in the bin average. The third component $w_{z,i}$ is the weighting factor based on the horizontally polarized

reflectivity. It is defined for each $\bar{x}_i$ as the azimuthally averaged reflectivity divided by the total sum of azimuthally averaged reflectivity values in the height bin. This last weighting step has been introduced previously in section 3.1.1 and is implemented to reduce the impact of low reflectivity values with low signal-to-noise ratio on the bin average.

The method therefore automatically puts more weight in measured volumes that most closely represent the contents of the height bin. This procedure allows the extraction of vertical profiles without artificial gaps regardless of moving average bin

height and distance to the PPI performing radar as data points not within the height bin are still considered as long as the range gate is intersected by the height bin. Depending on the distance to and the size of the chosen segment, the number of averaged data points per height bin can be similar or even higher than for QVPs (Bukovčić et al., 2020). As stated previously for the RHI scans, $Z_{dr}$ is calculated after the extraction process from extracted profiles of vertically and horizontally polarized radar

reflectivity.

The position of the defined segment can be chosen arbitrarily within the area covered by the operational PPI scans at locations of interest, following a given path (Murphy et al., 2020) or like in this study centered at a vertically pointing cloud radar. Similarly the size of the segment in azimuth and radial range can be varied freely. This is of exceptional use for the observation of strongly localized weather events like convective cells which require a small segment for accurate analysis or for the study of homogeneous weather situations like stratiform precipitation which can be examined over a wide area with high statistical accuracy.

**Composites of multiple radars**

As previously seen in Fig. 1, the German operational radar network features rather dense coverage of Germany where several radars have significant overlap with each other. This is especially true for the six PPI scans conducted at the lowest elevation angles and therefore with the highest spatial coverage.

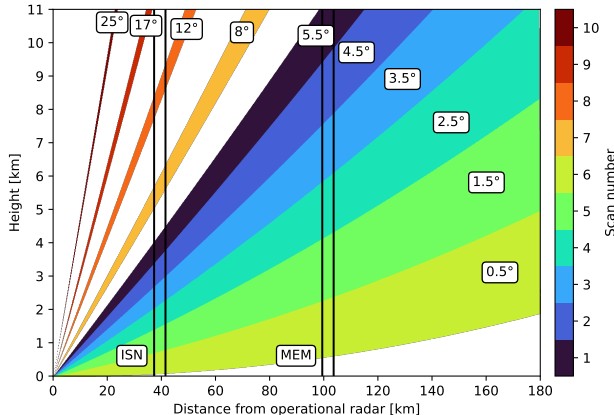

**Figure 7.** Scan strategy of the German operational radar network serviced by the DWD. Schematically shown are the ten PPI scans colored in order of execution with their respective elevation angles considering beam-broadening, earth curvature and refraction in earth's atmosphere. Black vertical lines mark the radial extent and position of the extracted BA-CVPS of ISN (at ca. 40 km) and MEM (at ca. 100 km) with regards to MIRA-35.

For higher signal-to-noise-ratios and additional data points it is therefore beneficial to try and combine the data of two or more operational radars in range of the point of interest. Since BA-CVPs consider beam-broadening, operational radars at different ranges from the point of interest can be included and are weighted according to beam extent. Additionally one operational radar can cover potentially occurring gaps in the data of another operational radar. This can be seen in Fig. 7 where the current scan strategy of the German operational radar network is depicted. Shown are the ten operationally performed PPI scans from a side view colored in order of execution with their respective elevation angle. Marked by black vertical lines is the radial extent

and range of the extracted BA-CVPs for ISN (at ca. 40 km) and MEM (at ca. 100 km) in relation to MIRA-35. While the extracted profile of ISN features higher vertical resolution at low altitude, where the data points are still spaced more narrowly, the profile also has noticeable gaps at higher altitudes. In contrast, the extracted profile of MEM has no gaps but only low vertical resolution as the data points are spaced further apart. The gaps in the profile of ISN are a result of the limited number of performed PPI scans and the short distance between ISN and MIRA-35 with little beam broadening. These gaps however can be covered with data of the extracted profile of MEM.

Combining two or more operational radars at different ranges to the point of interest therefore allows the filling of potentially occurring gaps caused by the operational PPI scan strategy and increases vertical resolution by consideration of more data points.

The main benefits of the BA-CVP method in comparison to the original CVP method by Murphy et al. (2020) can be summarized in the following bullet points:

– BA-CVP method considers beam broadening and weighs data points by area of intersection between height bin and radar beam instead of vertical distance to height bin used in the original CVP method

– Data points not within a height bin still contribute to the height bin average if measurement volume intersects the height bin, reducing gaps in areas with low data point density

– Possibility to combine the data of multiple radars at different distances to the point of interest due to beam-aware treatment of data points, increasing statistical significance and reducing gaps

– Gaps in extracted BA-CVPs are real, unmeasured gaps that were not part of any radar measurement volume.

## 3.2 Gaseous attenuation

Atmospheric gases can cause significant attenuation of radar signals especially at higher frequency like the Ka-band or when the radar signals have to travel over long distances in the lower, more dense atmosphere. For all involved radars and their measured reflectivities a gaseous attenuation correction is therefore implemented (ITU, 2019). The correction involves all oxygen and water vapor absorption lines where attenuation is expected to be relevant. The formula uses atmospheric pressure, relative humidity and temperature which is provided on a bi-daily basis (noon and midnight) by an atmospheric sounding station in Oberschleißheim roughly 13 km north of the position of MIRA-35. The sounding data is provided by the University of Wyoming sourced from the Deutscher Wetterdienst (DWD, 2019, 2021). For radar data between the bi-daily sounding measurements, the atmospheric attributes are interpolated with a non-overshooting cubic interpolation to correctly consider diurnal changes.

## 3.3 Hydrometeor attenuation

The presence of hydrometeors in the radar beam path can lead to significant attenuation in the returned reflectivity signal of weather radars. This is especially true for measurements over long distances but also for measurements where the beam re-

mains in or below the melting layer (e.g. at low elevation angles) and therefore encounters mostly liquid hydrometeors which inherently have a higher impact on attenuation. Since the radar beams of spatially separated radars follow different paths, accumulated attenuation can differ significantly and strongly influence dual wavelength ratio (DWR) calculations if the reflectivity measurements of two radars at different wavelengths are combined. In the case of spheroid hydrometeors this effect also becomes of importance for polarimetric variables like $Z_{dr}$ as different attenuation in the two polarization components directly affect and change measured $Z_{dr}$ values. The effects of hydrometeor attenuation on column-extracting methods like QVPs and CVPs has not been studied in detail so far. It is out of the scope of this paper to offer a comprehensive solution for the correction of hydrometeor attenuation regarding reflectivity measurements or differential attenuation relevant for $Z_{dr}$. To evaluate the feasibility to complement cloud radars with operational weather radars and all methods similar to BA-CVPs it is however still inevitable to consider occurring beam effects like hydrometeor attenuation qualitatively.

Most polarimetric radars are able to measure $\Phi_{dp}$ which has been successfully used to quantify hydrometeor attenuation (Bringi et al., 2001). $\Phi_{dp}$ however usually shows particularly high noise when compared to other radar moments and even other polarimetric quantities, especially when measured within operational measurement schemes. Although BA-CVPs utilize azimuthal averaging to increase statistical significance, simply relying on measured $\Phi_{dp}$ values as a marker for high hydrometeor attenuation to filter out affected $Z_{dr}$ and radar reflectivity ($Z_e$) values might not always be possible due to residual noise. To still be able to get at least an estimate about the accumulated attenuation caused by liquid hydrometeors a gate-by-gate approach is utilized. The method initially introduced by Jacobi and Heistermann (2016) solely relies on measured reflectivity values and is readily available in the Python package wradlib (Heistermann et al., 2013). In this paper no hydrometeor attenuation is applied to the measurements. Instead, the calculated liquid hydrometeor attenuation based on the gate-by-gate approach of Jacobi and Heistermann (2016) is compared to the extracted BA-CVPs of $\Phi_{dp}$ and the usage of the $\Phi_{dp}$ BA-CVPs as a qualitative marker to filter out measurement pixels with high attenuation is discussed.

## 4 Results

The newly developed method of BA-CVPs is applied to two different case studies. For each case, vertical profiles extracted from a dedicated RHI scanning radar (POLDIRAD or MHP) are compared to BA-CVPs extracted from the data provided by the two operational radars ISN and MEM. All vertical profiles are compared to the high resolution profiles measured by the vertically pointing MIRA-35. The comparison between dedicated and operational radar data aims to evaluate the feasibility to complement cloud radars with operational radar data.

### 4.1 Case study 1: 28 May 2019

The results of the newly introduced BA-CVPs are first presented for radar measurements collected on 28 May 2019 between 05:30 UTC and 13:50 UTC. On 28 May 2019, Germany was affected by the front side of a through extending southwards over France. Lifting at the front side caused precipitation in varying intensity with strong downpours happening occasionally over the whole observed time frame.

During measurements the dedicated radar POLDIRAD was facing towards MIRA-35 and doing two consecutive RHI scans every 10 minutes with an elevation angle velocity of $1°\mathrm{s}^{-1}$ for a total number of 100 scans.

Figure 8 shows all vertical profiles of $Z_\mathrm{e}$ in a time versus height manner collected during the specified time frame. For comparison, vertical profiles of MIRA-35 are depicted in 8 (a) and extracted vertical profiles based on RHI scans of the dedicated radar POLDIRAD are plotted in 8 (b). The BA-CVPs are shown in 8 (c) for ISN and 8 (d) for MEM, respectively. A composite for the data collected by the two operational radars is portrayed in 8 (e). Additional radar variables of interest like the mean Doppler velocity (MDV) in Fig. 9 (a) or the linear depolarisation ratio (LDR) in Fig. 9 (c) of MIRA-35 in comparison to the differential reflectivity ($Z_\mathrm{dr}$, Fig. 9 (b)) of POLDIRAD and the operational radars ISN and MEM Fig. 9 (d) are plotted in Fig. 9 for the specified time frame. Some profiles in both figures are masked since here either MIRA-35 was not pointing vertically and performing RHI scans or POLDIRAD was not pointing towards MIRA-35.

As can be seen in Fig. 8 (a) and (b) the 28[th] of May featured continuous liquid precipitation at the surface with varying intensity over the observed time range. Most precipitation fell in the form of stratiform rain. The stratiform precipitation was occasionally interrupted by convective cells moving over MIRA-35 most notably recognizable at 08:30 UTC, shortly after 10:00 UTC and 13:30 UTC with stronger rain intensity identifiable by high $Z_\mathrm{e}$ values of 30 dBZ for MIRA-35 or 40 dBZ for POLDIRAD. The first convective cell at 08:30 UTC also shows a very distinctive signature in the MDV of MIRA-35 in Fig. 9 (a).

These general features of precipitation can be seen in the extracted BA-CVPs of ISN and MEM in Fig. 8 (c) and (d) as well. All C-band radars show similar intensities in the stratiform and convective parts. However due to the longer distance to MIRA-35, the measurements of the operational radar MEM exhibit a reduced sensitivity with minimum measured values of around 0 dBZ compared to ISN and POLDIRAD with reflectivities down to -15 dBZ. The low reflectivity values above 7 km seen in Fig. 8 (a) of MIRA-35 are only partly reproduced by POLDIRAD and are completely missing for the two operational radars. This is again mostly due to the reduction of sensitivity with increasing range but also due to the different wavelengths that MIRA-35 and the C-band radars are operating at. Very small ice particles dominant at high altitudes are only detectable by high-frequency radars. In the case of ISN, gaps in the profiles stemming from the limited number of measured elevation angles are also observed as previously seen in Fig. 7. To fill these gaps, the BA-CVPs of ISN and MEM are combined into a composite which, as seen in Fig. 8 (e), offers mostly gapless coverage of the atmosphere up to 7 km in height for this case. Most details of the composite originate from the higher resolution data of the closer radar ISN. The data of MEM only supplements and fills gaps where necessary since only the lower three elevation angles can contribute.

The melting layer (ML) was situated at an approximate altitude of 2.4 km with little variability during the day and only a slight decrease in height to roughly 2 km during the second half of the observed time frame as identifiable by the sharp increases in reflectivity at those heights seen in Fig. 8 (a) for MIRA-35 and as a bright band for POLDIRAD in Fig. 8 (b). The height of the melting layer can also be identified using the $Z_\mathrm{dr}$ measurements of POLDIRAD in Fig. 9 (b) which shows a vertically shallow bright band between 2.3 km and 2.6 km. This is confirmed by the MDV and the LDR measurements in Fig. 9 (a) and (c). The MDV shows a very concise transition from low falling speeds just below 0 $\mathrm{ms}^{-1}$ to high falling speeds at -5 $\mathrm{ms}^{-1}$ while the LDR displays a very shallow bright band with elevated LDR values up to -10 dB.

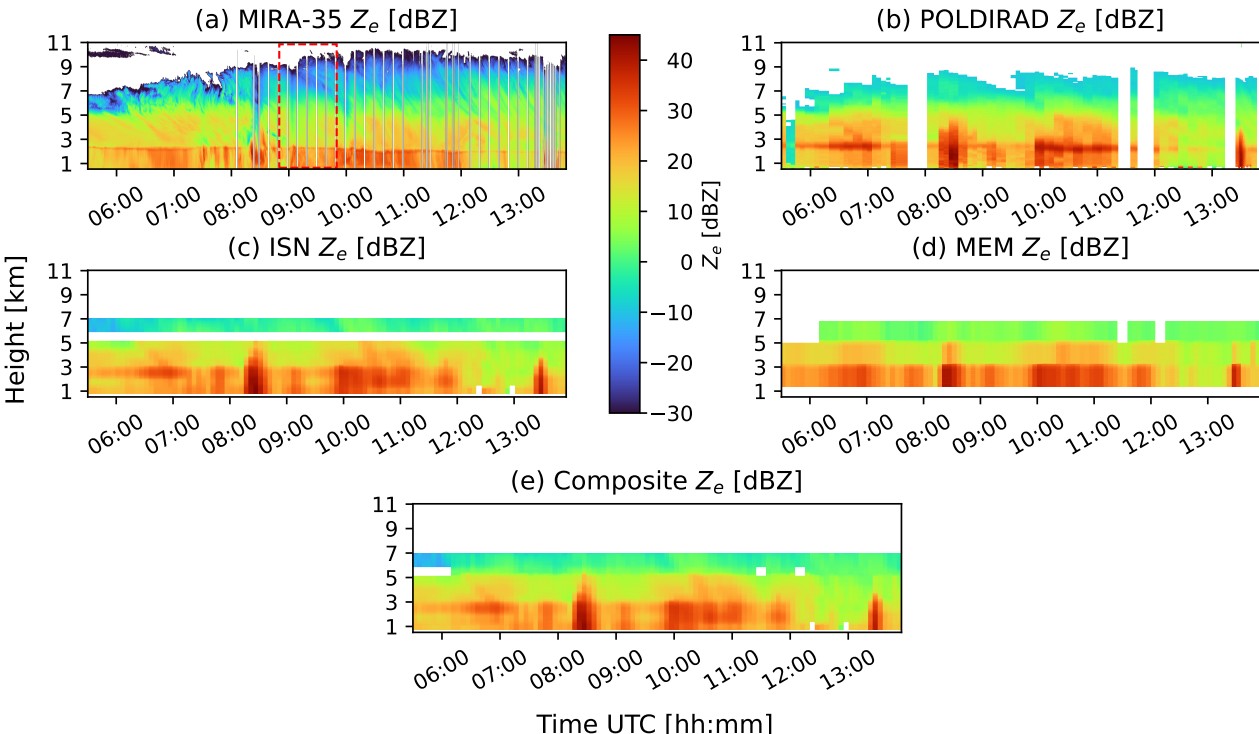

**Figure 8.** Measured radar reflectivity in vertical profiles of MIRA-35 (a), in extracted vertical profiles based on RHI scans of POLDIRAD (b), in BA-CVPs of ISN (c) and MEM (d) and as composite of ISN and MEM (e) collected on 28 May 2019 between 05:30 UTC and 13:50 UTC. The area enclosed by the red rectangle in (a) will later be used to calculate averaged vertical profiles.

Similar to the measurements of MIRA-35 and POLDIRAD, the data of the two operational radars offers the possibility to determine the height of the ML. Its signature can be observed between 2.0 km and 2.7 km by the strong increase in reflectivity seen in Fig. 8 (e). This value roughly resembles the previously estimated values of 2.4 km for the first part of the day and 2 km for the second part of the day based on the high resolution data of MIRA-35 and POLDIRAD. However, it is obvious

that the signal of the very shallow ML is not resolved quite as well as with MIRA-35 or POLDIRAD since fewer elevation angles and range gates are available. This becomes especially apparent when one tries to estimate the ML height only from the measurements of radar MEM. As stated previously only the lower three radar beams of MEM contribute to the BA-CVPs and because of significant beam broadening all features are broadened as well. Alone from the data of the radar MEM, the ML height can, therefore, only be estimated to lie between 1.2 km and 3 km within the lowest measured radar beam.

The measurements of $Z_{dr}$ of the two operational radars as seen in Fig. 9 (d) can partly be used to confirm these findings. The definition of a locally limited and sharp increase in $Z_{dr}$ as previously seen for the $Z_e$ of MIRA-35 or the $Z_{dr}$ of POLDIRAD in low heights however is generally not possible. For this specific case, the ML had a height that was located directly between two consecutive beams for the closer radar ISN and still within the first elevation of radar MEM. Depending on local and temporal

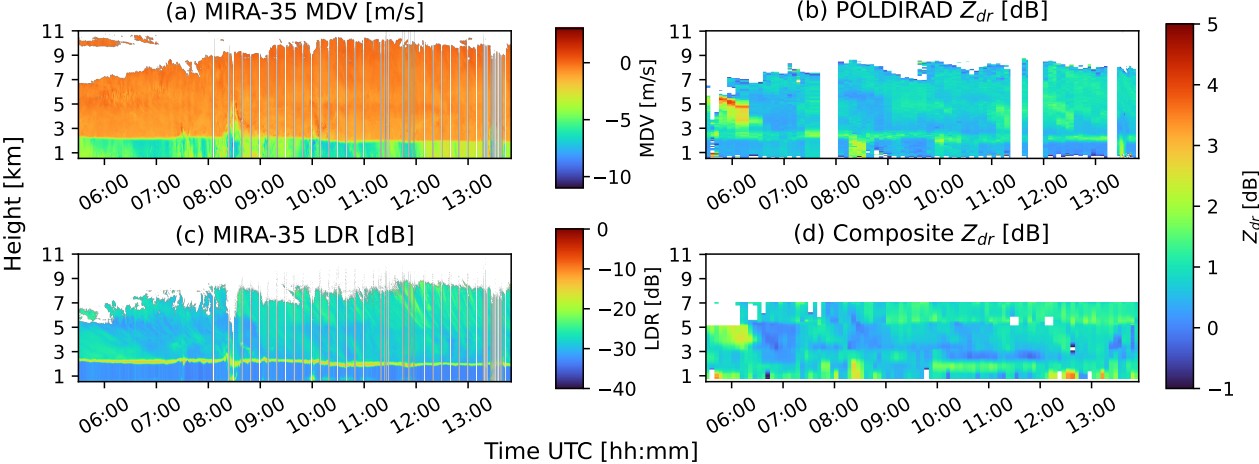

**Figure 9.** Measured mean Doppler velocity (a) and linear depolarisation ratio (c) of MIRA-35 in vertical profiles, differential reflectivity in extracted vertical profiles based on RHI scans of POLDIRAD (b) and differential reflectivity in a BA-CVP composite of ISN and MEM (d) for data collected on 28 May 2019 between 05:30 UTC and 13:50 UTC.

fluctuations in the height of the ML, either the lower or upper beam of radar ISN was able to detect the increase in $Z_{dr}$. This
for example becomes particularly evident, when the two time frames between 07:30 UTC to 09:30 UTC and 10:00 UTC to 12:00 UTC are compared. In the first time frame, the radar beam with the higher elevation captures the signal of the ML and shows higher $Z_{dr}$ values for a ML height of around 2.2 km to 2.7 km. In the second time frame after a slight decrease in the ML height shortly before 10:00 UTC, the lower radar beam is able to measure the signal of the ML. The ML height can then be estimated to be between 1.6 km and 2.2 km in height. For both time frames, the previously determined values based on high
resolution data of POLDIRAD and MIRA-35 as well as the estimates from the $Z_e$ measurements of the operational radars lie within these ranges.

The $Z_{dr}$ measurements of POLDIRAD in general show mostly positive values above the ML indicating the existence of horizontally aligned particles. This is also the case for the operational composite of $Z_{dr}$ in Fig. 9 (d). Finer details like temporary
increases in $Z_{dr}$ as seen between 07:10 UTC to 08:10 UTC or 09:00 to 10:10 UTC at heights of 3 km to 6 km can be identified for POLDIRAD and the operational radars with similar intensity. At around 06:00 UTC a strong increase in $Z_{dr}$ at altitudes between 4 km and 5 km can be identified in the measurements of POLDIRAD. The operational composite manages to reproduce this increase and because of the slightly different measurements times (operational: 10 PPI scans during 5 minutes, dedicated: 2 consecutive RHI scans every 10 minutes) even manages to add slightly more detail in the internal structure of the feature.
Figure 10 shows extracted BA-CVPs of measured differential phase ($\Phi_{dp}$) in comparison to the BA-CVPs of calculated liquid hydrometeor attenuation ($A_{hyd}$) following the gate-by-gate approach of Jacobi and Heistermann (2016) for all involved C-band

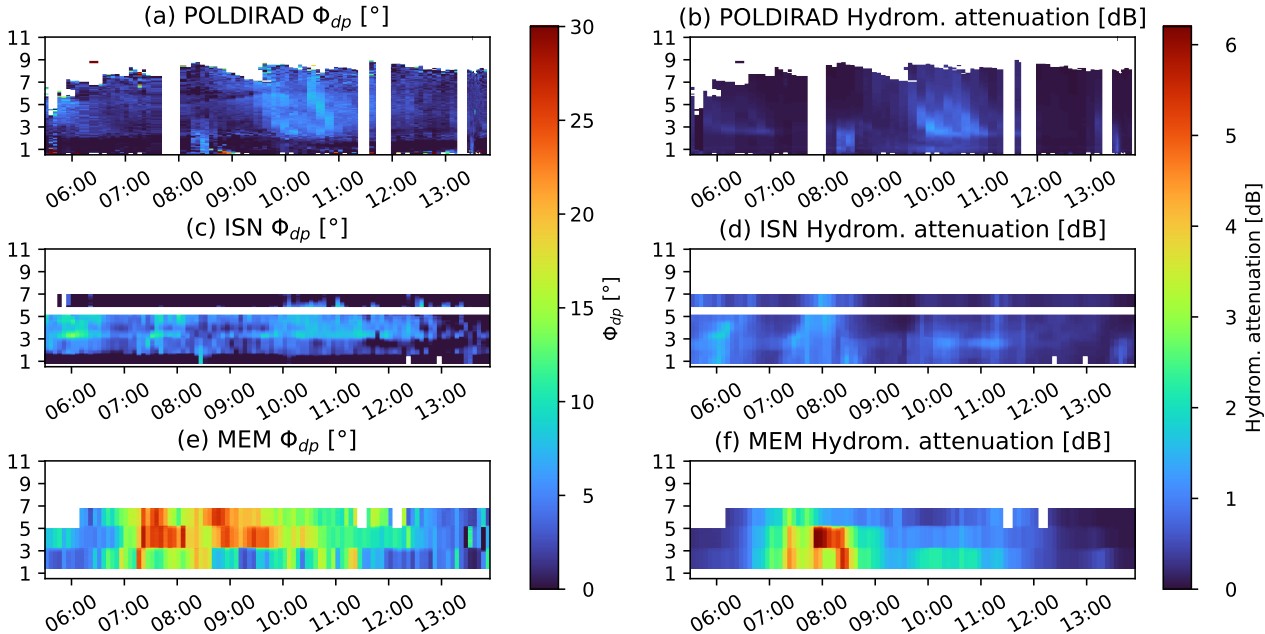

**Figure 10.** Extracted BA-CVPs of differential phase in comparison to the calculated liquid hydrometeor attenuation for POLDIRAD in (a) and (b), for ISN in (c) and (d) and for MEM in (e) and (f) for data collected on 28 May 2019 between 05:30 UTC and 13:50 UTC.

radars as previously introduced in section 3.3. Since $\Phi_{\mathrm{dp}}$ as well as $A_{\mathrm{hyd}}$ are both affected by different beam propagation, a composite containing the data of both operational radars is not applicable. The lowest elevation angles are often influenced 465 by clutter and therefore little to no reliable signature of hydrometeors for lower altitudes with range gates close to the radar is measurable.

As seen in Fig. 10 (a), POLDIRAD as the closest radar to MIRA-35 generally experiences the least amount of attenuation with $\Phi_{\mathrm{dp}}$ values staying below 10 degrees. Events with slight increases in $\Phi_{\mathrm{dp}}$ indicating the presence of strongly attenuating hydrometeors in the beam paths from 05:45 UTC to 07:00 UTC, around 0815 UTC and again from 09:30 UTC to 11:00 UTC 470 can also be identified in the plot of $A_{\mathrm{hyd}}$ reaching maximum values of around 1.5 dB. In general however, attenuation of POLDIRAD signals remained rather low for the entire measurement time frame.

Similarly, the second furthest radar to MIRA-35, ISN, also showed events with only small increases in $\Phi_{\mathrm{dp}}$ ranging in time from 05:30 UTC to 06:45 UTC, from 07:15 UTC to 08:30 UTC and from 09:30 UTC to 12:30 UTC as seen in Fig. 10 (c). These events showed a slightly higher $\Phi_{\mathrm{dp}}$ when compared to POLDIRAD but generally stayed below values of 15 degrees. 475 The features of $\Phi_{\mathrm{dp}}$ are again also identifiable in the calculated hydrometeor attenuation for ISN shown in Fig. 10 (d) at similar times. Compared to POLDIRAD, $A_{\mathrm{hyd}}$ experienced by ISN is generally a bit higher reaching maximum values of around 2 dB. Reasons for this are discussed in section 5.

MEM shows the highest increases in $\Phi_{\mathrm{dp}}$ reaching values of up to roughly 30 degrees as seen in Fig. 10 (e). One consecutive

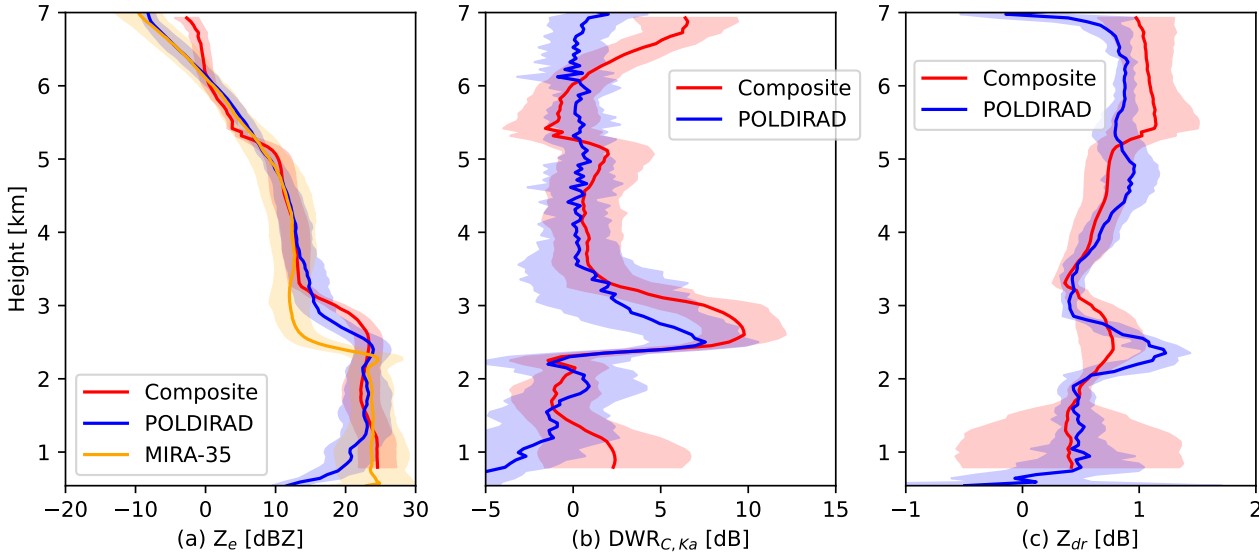

**Figure 11.** Averaged profiles of $Z_e$ in (a), $DWR_{C,Ka}$ in(b) and $Z_{dr}$ in (c) for the stratiform time period on 25 May 2019 between 08:50 UTC and 09:50 UTC (marked in Fig. 8 (a) as red rectangle). The colored area is the standard deviation.

event with elevated values of $\Phi_{dp}$ ranging in time from 07:00 UTC to 10:00 UTC can be identified. In this case the calculated
hydrometeor attenuation for MEM in Fig. 10 (f) is not able to fully capture this event. Only in the beginning between 0700
UTC and 08:45 UTC, strongly increased hydrometeor attenuation is seen with values of up to 6.2 dBZ. During the time from
08:45 UTC to 10:00 UTC only smaller values of $A_{hyd}$ between 1 dB to 2.5 dB are present contradicting the strong increase in
$\Phi_{dp}$ during that time.

In order to further analyze the performance of the BA-CVP method, vertical profiles averaged over the time period between
08:50 UTC and 09:50 UTC were calculated. This time period is marked as a red rectangle in Fig. 8 (a). Since the 28 May
2019 consists of precipitation in varying intensity with several convective cells traveling over the region of interest and also
through the beams of each radar, a more stratiform region with negligible hydrometeor attenuation as expected from Fig. 10
for all involved radars was chosen for best comparability. The results are shown in Fig. 11 for $Z_e$, the dual-wavelength ratio
($DWR_{C,Ka}$) between MIRA-35 and the C-band radars as well as for $Z_{dr}$.
The averages of the extracted composite BA-CVPs based on the operational radars generally show good agreement with the
averages of the virtual profiles based on POLDIRAD for all three radar variables. Only in areas where the composite is either
based on just one radar beam (roughly above 6 km height) or might be affected by clutter (below 1.5 km height), higher
deviations are visible. Additionally, the radar signal of the ML in both $Z_e$ and $Z_{dr}$ appears a bit more smeared out for the
average of the operational composite as expected from previous observations. Specifically the ML peak in $Z_{dr}$ is less intense.

## 4.2 Case study 2: 08 July 2021

As another example for the application of BA-CVPs a second case study with data measured on 08 July 2021 from 16:58 UTC to 18:01 UTC is presented.

The measurements used in this case study were collected by the dedicated radar MHP which was pointing towards MIRA-35 and performing one RHI scan every two minutes for a total of 32 scans. As MHP is about 2.5 times further away than POLDIRAD, the results of the dedicated measurements are expected to be affected by beam-broadening which coarsens the resolution.

On the 08 July 2021 Europe was under the influence of a trough starting in front of the south-eastern coast of Norway spanning over the North Sea down to the Côte d'Azur. At the front side, moist and warm air masses flowing from south to north facilitated growth of strong convective systems by orographic lifting in the Alpine region. From here thunderstorms developed which traveled northeastwards through the measurement region.

Figure 12 contains all vertical profiles of $Z_e$ collected during the specified time frame. In Fig. 12 (a) the vertical profiles measured by MIRA-35 are shown while in Fig. 12 (b) the vertical profiles extracted from RHI scans of the dedicated radar MHP are depicted. Figure 12 (a) and (b) serve as a comparison for the extracted BA-CVPs of ISN and MEM which can be seen in Fig. 12 (c) and Fig. 12 (d). The composite BA-CVPs based on data from both operational radars is displayed in Fig. 12 (e). Additional radar variables are presented in Fig. 13. MDV and LDR measurements of MIRA-35 are depicted in Fig. 13 (a) and Fig. 13 (c) while a comparison between the measured $Z_{dr}$ of MHP to the BA-CVP composite of ISN and MEM is shown in Fig. 13 (b) and Fig. 13 (d).

This case consists mostly of stratiform precipitation with stronger precipitation events taking place at the beginning of the observed time frame between 17:00 UTC and 17:15 UTC as well as from 17:30 UTC to 17:50 UTC as seen by elevated values of $Z_e$ of around 25 dBZ below the ML for MIRA-35 and MHP in Fig. 12 (a) and (b). A very brief period without surface precipitation exists from shortly after 17:15 UTC to 17:20 UTC. The reflectivity measurements of ISN and MEM in Fig. 12 (c) and (d) manage to reproduce all general features previously seen in the measurements of MIRA-35 and MHP albeit at lower resolution. On this date, more high ranging clouds upwards of 7 km were present, marking the region where the closer radar ISN has less coverage due to the limited number of measured elevation angles. As seen in Fig. 12 (e) a more accurate picture of the cloud situation can therefore only be achieved when the BA-CVPs of both operational radars are combined into a composite. The resulting composite possesses only a limited amount of small gaps and manages to reproduce the measurements of the dedicated radar MHP in good quality albeit the reduced time resolution. Because of the greater distance to MIRA-35 or hydrometeor attenuation, the measurements of MEM have less sensitivity and therefore fail to capture the low reflectivity values present at very high altitudes just below 11 km and at the very end of the measurement time frame.

The ML can be identified in Fig. 12 (a) at an altitude of approximately 3 km with little variability during the measurement time based on the strong increase in $Z_e$ at that height. The height of the ML is also confirmed by measurements of MDV and LDR in Fig. 13 (a) and (c). MDV shows a clear and concise transition from low fall speeds at around 0 ms$^{-1}$ to values at

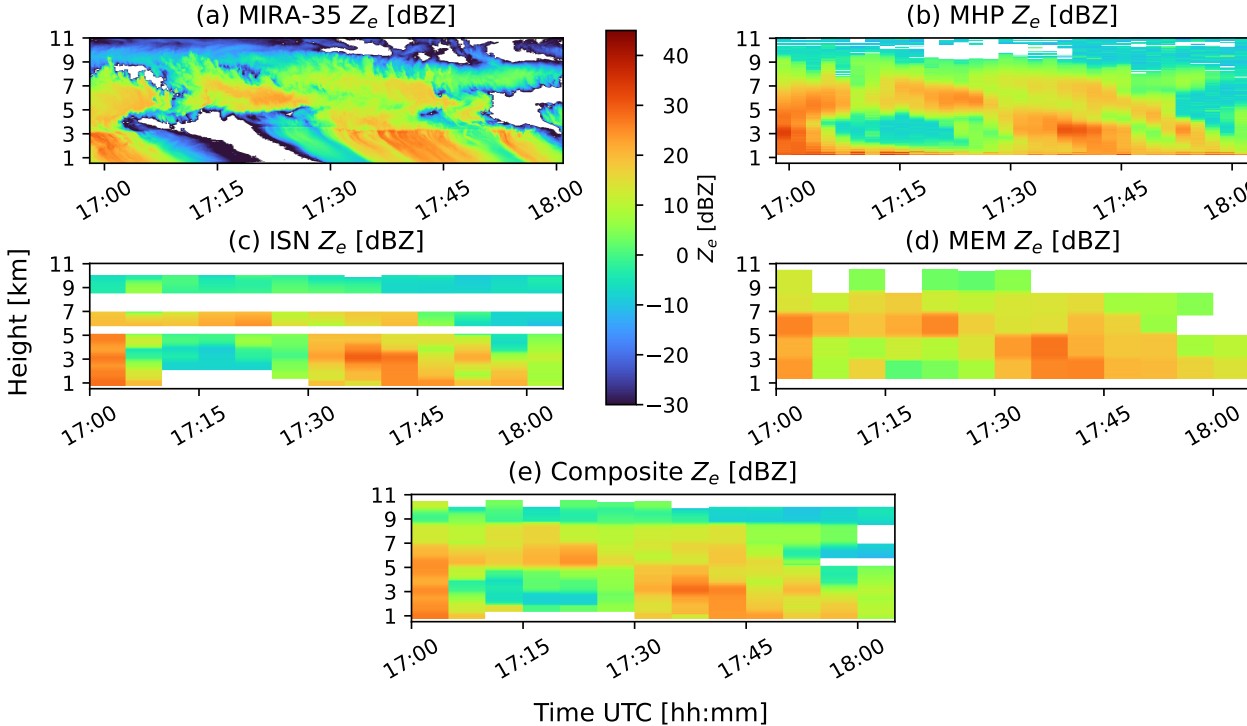

**Figure 12.** Measured radar reflectivity in vertical profiles of MIRA-35 (a), in extracted vertical profiles based on RHI scans of MHP (b), in BA-CVPs of ISN (c) and MEM (d) and as composite of ISN and MEM (e) collected on 08 July 2021 between 16:58 UTC and 18:01 UTC.

around -5 ms$^{-1}$ while LDR displays a vertically very shallow bright band with LDR values between -10 dB to -20 dB. In
contrast, the difference in resolution between MIRA-35 and MHP and the presence of clouds with high reflectivity above and
around the height of the ML as well as significant variance in the precipitation intensity, complicate the determination of the
ML height solely based on measurements by the radar MHP. Very subtle increases in $Z_e$ shortly before 17:00 UTC, between
17:38 UTC and 17:45 UTC as well as at around 17:54 UTC at heights of about 3 km hint to the position of the ML; they are
however, difficult to interpret without the measurements of MIRA-35. Similarly, no clear estimation of the ML height based on
the operational data is possible because of the previously mentioned complexity in precipitation intensity and cloud structure.
The composite of the operational radars for $Z_{dr}$ follows a similar form and generally features similar values of $Z_{dr}$ as the
dedicated measurements. Especially the strong increase at the beginning of the measurement time frame at a height of roughly
7 km is visible in both datasets. Other fine details present in the measurements of the dedicated radar are however hard to
identify in the composite of the operational radars. This is mostly due to the reduced time resolution of the operational scans
but also due to the extended time periods with low reflectivity where reliable $Z_{dr}$ measurements are not possible and therefore
lead to larger gaps in the composite.

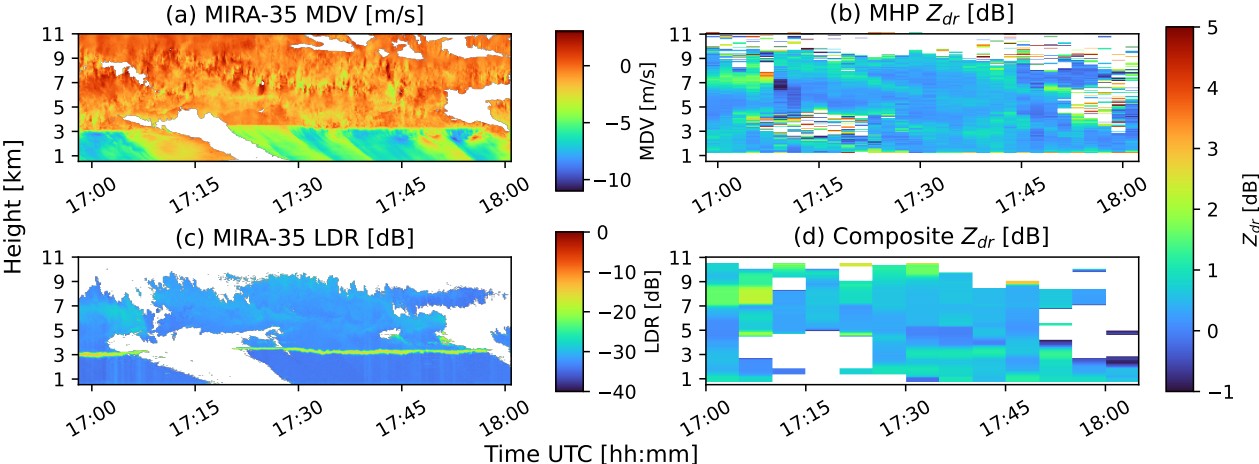

**Figure 13.** Measured mean Doppler velocity (a) and linear depolarisation ratio (c) of MIRA-35 in vertical profiles, differential reflectivity in extracted vertical profiles based on RHI scans of MHP (b) and differential reflectivity in a BA-CVP composite of ISN and MEM (d) for data collected on 08 July 2021 between 16:58 UTC and 18:01 UTC.

## 5    Discussion

In the original columnar vertical profile method by Murphy et al. (2020) the occurrence of gaps in extracted profiles at locations far from the involved radar or based on measurements with sparse elevation scans proved to be a major limitation. The size
of the gaps in this case was related to the bin size (Cressman radius of influence) chosen by the authors during the regular grid averaging step. Where this bin size is smaller than the vertical distance between PPI scans, gaps are inevitable. As the beam-aware columnar vertical profile method follows a beam-aware approach, all measured radar beams are considered with their real physical extent. Consecutive elevation scans performed at elevation angles that are separated by exactly the distance of the beamwidth, therefore will never show gaps in the extracted profiles regardless of radar distance and chosen averaging
bin size. The beam-aware approach also allows to cover gaps which are the result of successive measurements at elevation angles that are further apart than the beamwidth by utilizing data of more than one operational radar. Furthermore and similar to the azimuthal averaging performed in both the original CVP method and the new BA-CVP method, the inclusion of data of additional radars improves the signal-to-noise-ratio. It is however important to note that most details present in the composites for the given two-radar combination in this paper rely on the closer radar.
The BA-CVP method therefore is ideally used with data of operational radar networks that offer only a limited number of measured elevation angles in their PPI scan strategy but show reasonable overlap of different radars at different distances to the points of interest. Since the position of the segments can be chosen arbitrarily, BA-CVPs can be extracted along a given route like the flight paths of planes or the footprints of satellites e.g. EarthCare (Illingworth et al., 2015). This has been done previously for an airplane collecting in-situ measurements by Murphy et al. (2020) but only locally within the range of one

radar. As the data of numerous radars is incorporated in BA-CVPs, the chosen path can cross several measurement areas of different radars and therefore be used within the total range covered by the operational radar network.

The extracted BA-CVPs presented in this work were based on data of the German radar network operated by the DWD whose scan strategy only offers ten different elevation angles. Nevertheless, the achieved results here are of similar quality and resolution when compared visually to the CVPs presented by Murphy et al. (2020) which were based on measurements of one radar of the American NEXRAD system operating in VCP-215 with a total of 15 measured elevation angles and therefore intrinsic higher height resolution.

The use of the newly implemented BA-CVP methodology on data provided by operational radars of the DWD revealed mixed but also promising results when compared to measurements of dedicated radars and paired with a vertically pointing cloud radar.

Features identifiable in $Z_{\mathrm{e}}$ measurements of the dedicated radars and the cloud radar were also identifiable in the BA-CVPs of the operational radars with similar intensity and acceptable resolution to complement vertically pointing cloud radars with polarimetry. Even limited height estimations for the shallow ML were possible although for an exact estimate one should always rely on the high resolution measurements of MDV and LDR by the cloud radar. Determining the ML height solely based on measurements by the C-band radars, both dedicated or operational ones, might not always be possible as became apparent in the second case study on 08 July 2021 where no clear ML was identifiable.

Extracted BA-CVPs of $Z_{\mathrm{dr}}$ showed reasonable similarities when compared to the measurements of the dedicated radars. Especially areas with elevated $Z_{\mathrm{dr}}$ values with sufficient length in time and height were well captured. Smaller details can often also be distinguished. Due to the different averaging steps, the composites showed reduced noise often comparable to the noise present in the measurements of the dedicated radars.

Another polarimetric quantity of interest is $\Phi_{\mathrm{dp}}$. Since $\Phi_{\mathrm{dp}}$ is cumulative along the beam, extracted BA-CVPs of $\Phi_{\mathrm{dp}}$ are a good indicator how much attenuating hydrometeors have been encountered on each beam path up until the point of interest at the vertically pointing cloud radar. Measurements of $\Phi_{\mathrm{dp}}$ by the dedicated radar POLDIRAD and the operational radar ISN showed good results with clear indications of time frames where higher or lower hydrometeor attenuation was expected. These findings were also reproduced by using a simple gate-by-gate calculation of liquid hydrometeor attenuation. At low elevation angles however, the measurements of $\Phi_{\mathrm{dp}}$ were strongly affected by clutter and no reliable measurements were possible. The measurements of $\Phi_{\mathrm{dp}}$ of the operational radar MEM showed only limited comparability to the gate-by-gate calculation of liquid hydrometeor attenuation. While examining the beam path for MEM during time periods where the extracted $\Phi_{\mathrm{dp}}$ BA-CVPs and the calculated liquid hydrometeor attenuation differ significantly (e.g. 08:30 UTC to 10:00 UTC), several cells with strong reflectivity signal traveling through the beam of MEM were observable. The influence of the melting layer of convective cells on the radar beams traveling within the melting layer at an oblique angle over a long distance might lead to effects that are captured by $\Phi_{dp}$ but not the reflectivity-based liquid hydrometer attenuation by Jacobi and Heistermann (2016). Additionally, although BA-CVPs rely on azimuthal averaging to increase the signal-to-noise-ratio, the size of the chosen segment considering the long distance of MEM to MIRA-35 might not be big enough to adequately reduce residual noise in $\Phi_{\mathrm{dp}}$ or $Z_{\mathrm{e}}$ as the data point density in the segment decreases with increasing distance. Nevertheless, the $\Phi_{\mathrm{dp}}$ measurements can be used as a

marker to filter out measurement pixels with high hydrometeor attenuation or, conversely, be used to confirm the legitimacy of combining radar measurements of spatially separated radars during time frames with low $\Phi_{dp}$ signal.

A key limitation which cannot be solved by BA-CVPs or any other method using operational radar data is the limited time resolution of operational scan strategies. Especially in the second case study, where the time resolution of the dedicated radar was roughly 3 times higher, finer details present in the dedicated measurements cannot be distinguished in the extracted BA-CVPs.

BA-CVPs are also limited by the number of contributing elevation angles of the operational radars and the size of the chosen segment which dictates the number of contributing range gates and azimuth angles. This can lead to a lower height resolution when compared to RD-QVPs and QVPs. The signal-to-noise-ratio however can be comparable or even better depending on the size of the chosen segment. Additionally, the choice of a local segment preserves spatial information and allows the study of small local events like convective cells traveling right over the point of interest away from the actual location of the operational

radar.

For future work utilizing the method of BA-CVPs, the decreasing sensitivity with range might also be an important factor to consider. While in the present study most details even with low reflectivity were captured by the closer operational radar ISN, the further radar MEM showed perceptible loss in sensitivity and therefore lack of low reflectivity values. For cloud radars far from any operational radar this might prove to be a major limitation.

Although the second case on 08 July 2021 benefited from a higher data point density due to the decrease in the range gate size of the raw data from 1000 m down to 250 m, no clear effect was visible. For a more thorough study of the influence of the data point density on the quality of BA-CVPs, similar time periods of the two different cases could be compared. However, this more in-depth analysis is beyond the scope of the present paper.

**Consequences for the German radar network**

To study future use cases of the BA-CVP method beyond the combination of the two radars ISN and MEM presented in the two case studies above and to identify potential locations suitable for the installation of additional vertically pointing cloud radars that could be paired with operational radar data, a systematic quality analysis of the radar coverage provided by the

operational radar network in Germany is carried out.

Ideally, a BA-CVP has high vertical resolution and no significant gaps without data. Additionally each height bin should contain as many data points as possible for a high signal-to-noise-ratio. High vertical resolution can be achieved by extracting BA-CVPs close to operational radars as previously seen for the radar ISN in Fig. 7. Additional radars, potentially further away like the radar MEM, can be used to cover occurring gaps. A higher number of contributing radars furthermore increases the

vertical resolution and signal-to-noise-ratio by supplying additional data points. The quality of a BA-CVP is therefore affected by a number of different influences like the distance to surrounding radars, the number of radars covering the area or the local topographical relief. We summarize these influences in the following three, more abstract factors: Vertical resolution, coverage of the atmosphere and a high signal-to-noise-ratio. To study the expected BA-CVP quality based on the national German radar network, the radar coverage with beam-aware and uniquely identifiable contributions of all 17 operational radars is interpolated

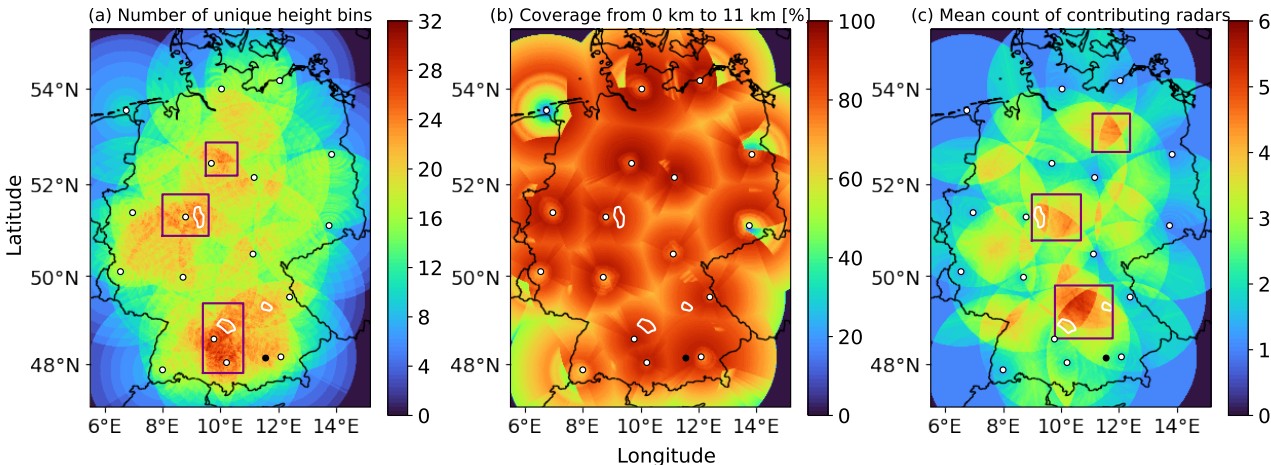

**Figure 14.** Depicted are three different scores to qualitatively rate the coverage of the operational radar network in Germany for use with BA-CVPs. In (a) the total number of unique height bins contained is shown. (b) reflects the total number of filled height bins in relation to the total number of available height bins and (c) details the number of contributing radars. Only heights between 0 km and 11 km are considered. The positions of all 17 operational radars are marked with a white dot. The black dot is the position of MIRA-35. Purple rectangles in (a) and (c) mark visually identifiable regions with either a high number of unique height bins (a) or mean number of contributing radars (c). Areas encased by a white line have ≥20 unique height bins, ≥80 % of all height bins are filled and at least three or more radars are contributing on average.

on a regular grid in Mercator projection by nearest-neighbor interpolation. The resolution of the grid is 2km×2km×0.1km between heights of 0 km to 11km.

To quantify the vertical resolution, the number of unique height bins for each geographic position is computed. The higher the number of unique height bins, the higher the expected vertical resolution of the BA-CVP if extracted at the location. A height bin is considered unique if it consists of a unique combination of elevation angles and operational radar contributions when

compared to all other height bins at the given position. The number of unique height bins for each geographical location is depicted in Fig. 14 (a). The coverage of the atmosphere is rated by computing the total number of filled height bins in relation to the total number of available height bins. A height bin is considered filled as long as at least one measured operational data point contributes to the height bin. The ratio of filled height bins is shown in Fig. 14 (b). And finally, as a marker for the statistical accuracy, the total number of contributing radars is computed for each height bin and then averaged for each geographic

position. The results can be seen in Fig. 14 (c). The position of the 17 operational radars is marked by white dots. The black dot is the position of MIRA-35 and therefore the region of main interest for the two case studies introduced previously.

As seen in Fig. 14 (b), the coverage of the atmosphere between 0 km and 11 km of the national German radar network usually ranges between 80% and 90% and only drops below 50% close to the border or outside of German territory. For the two most eastern radars and the radar in the north-western corner, coverage close to the radars is low since here the radar beams are

not yet broadened and more distant radars that could provide coverage are out of range. In some parts of Germany, the radar

beams are occasionally blocked by mountain ranges hence reducing the ratio of filled height bins. This is especially true for the southern radars where the measurements are hindered by the Alps.

In contrast to the generally high ratio of filled height bins, the number of unique height bins strongly depends on the geographic location. The number of unique height bins in Fig. 14 (a) usually remains between 16 and 20 for most locations in Germany. Again close to the borders, but especially in the north-eastern corner and outside of German territory lower values can be found. Three bigger areas with an increased number of the unique height bins ranging between 24 to 30 are visually identifiable and marked with purple rectangles in Fig. 14 (a).

Similarly to the number of unique height bins, the mean number of contributing radars differs strongly for different locations in Germany. Most areas are covered by two or even three radars. Close to the borders however or outside of Germany, contributions of only one radar are common. Three bigger areas where 4 or more radars contribute on average are marked with purple rectangles in Fig. 14 (c).

Ideal locations for the extraction of BA-CVPs based on operational radar data of the German radar network have a high number of unique height bins, good atmospheric coverage with no gaps and a high number of contributing operational radars. Areas encased by a white line in Fig. 14 (a), (b) and (c) fulfill these criteria having $\geq 20$ unique height bins, $\geq 80$ % percentage coverage of the atmosphere between 0 km and 11 km and at least three or more radars contributing on average. These areas therefore pose as ideal candidates for the future installation of additional vertically pointing measurement equipment since it can easily be complemented by the measurements of the national German radar network.

Figure 14 reveals that the location of MIRA-35 which was used as point of interest in this study is not part of the ideal regions. While featuring average coverage of the atmosphere at 85 % and a slightly above average number of unique height bins at around 20, the mean number of contributing radars is 2.5. The location of MIRA-35 therefore represents a good example for the average situation that is encountered in Germany when BA-CVPs are to be extracted based on data provided by the national German radar network. This and the good results achieved at the location of MIRA-35 with only two instead of all available radars in range in the two case studies presented before encourage the usage of BA-CVPs in whole Germany to further augment vertically pointing measurement equipment.

## 6  Conclusions

The present study examined the synergistic use of operational C-band weather radars operated by the DWD and a vertically pointing cloud radar in the Ka-band by analyzing two different case studies. For this purpose the original CVP approach was developed into the BA-CVP method by introducing several advancements. BA-CVPs preserve the initial benefits like arbitrary choice of the segment location and increased signal-to-noise-ratio by azimuthal averaging of the chosen segment but also allow the consideration of beamwidth and beam-broadening during propagation which represents a significant improvement. This made it possible to incorporate the data of 2 operational radars in range of the vertically pointing cloud radar correctly weighted by their distance for increased data point density and signal-to-noise-ratio in composites resulting in fewer gaps and more usability of the method over the total operational radar range.

Utilizing the newly introduced BA-CVP methodology the pairing of operational weather radar data focusing on radar polarimetry from an oblique perspective with the high-resolution measurements of a vertically pointing cloud radar were discussed. The extracted BA-CVPs based on operational radars were compared to extracted vertical profiles based on data collected by dedicated RHI scan performing radars pointing towards the cloud radar in two different case studies. The BA-CVPs showed good agreement in $Z_e$ and $Z_{dr}$ being able to resolve even fine details like the ML in some cases. In addition increased values of $\Phi_{dp}$ were evaluated as a marker for strong hydrometeor attenuation on the beam path and compared to liquid hydrometeor attenuation calculations based on a gate-by-gate approach showing reasonable similarities and good usability to classify time frames and/or vertical bins with strong hydrometeor attenuation. The combination of more than one operational radar in the BA-CVP method proved to be crucial for gapless coverage of the atmosphere with the chosen segment size and the limited amount of operational measurements.

As a key limitation of the BA-CVP method and any other method sourcing data from weather radars, the low time resolution as consequence of the scanning scheme used in operational radar networks was identified.

For the future use of BA-CVPs based on operational radar data, a quality analysis of the radar coverage provided by the national German operational radar network was conducted. The German radar network in general provides good coverage over the whole Germany. The expected vertical resolution for BA-CVPs as well as the mean number of contributing radars however depend on the geographical location of the point of interest. Three different locations with the best compromise between vertical resolution, atmospheric coverage and high number of contributing radars were identified and could be used for the installation of additional vertically pointing cloud radars. The location of MIRA-35 used as point of interest in the two case studies presented in this work, turned out to be a good example for the average situation encountered in Germany if BA-CVPs are to be extracted. The promising results achieved with BA-CVPs at the position of MIRA-35 encourage further use of BA-CVPs based on operational radar data in combination with other cloud radars at different locations or radar carrying satellites like EarthCARE (Illingworth et al., 2015). Future studies utilizing the combination of operational radar BA-CVPs with vertically pointing cloud radars could overcome observational limitations and implement a radar-driven retrieval able to constrain mass, size and shape of ice hydrometeors utilizing polarimetric radar variables in combination with Doppler fall-speed velocity and dual-wavelength ratio measurements.

**Appendix A:  Measurement strategy of the national German radar network operated by the DWD**

**Table A1.** Overview over radar parameters for operational PPI scans performed by the national German radar network operated by the DWD. Two values for the PRF indicate PRF staggering.

| Parameter | PPI 0 | PPI 1 | PPI 2 | PPI 3 | PPI 4 | PPI 5 | PPI 6 | PPI 7 | PPI 8 | PPI 9 |
|---|---|---|---|---|---|---|---|---|---|---|
| Elevation angle (°) | 5.5 | 4.5 | 3.5 | 2.5 | 1.5 | 0.5 | 8 | 12 | 17 | 25 |
| Azimuth speed (°s$^{-1}$) | 16 | 16 | 16 | 16 | 16 | 12 | 18 | 30 | 30 | 30 |
| PRF (Hz) | 600/800 | 600/800 | 600/800 | 600/800 | 600/800 | 600/800 | 800/1200 | 2410 | 2410 | 2410 |
| Pulse width (µs) | 0.8 | 0.8 | 0.8 | 0.8 | 0.8 | 0.8 | 0.8 | 0.4 | 0.4 | 0.4 |
| Range resolution (m) | 1000[*] 250[†] | 1000[*] 250[†] | 1000[*] 250[†] | 1000[*] 250[†] | 1000[*] 250[†] | 1000[*] 250[†] | 1000[*] 250[†] | 1000[*] 250[†] | 1000[*] 250[†] | 1000[*] 250[†] |
| Range (km) | 180 | 180 | 180 | 180 | 180 | 180 | 124 | 60 | 60 | 60 |

[*] 28th May 2019; [†] 8th July 2021

*Code availability.* The software code can be made available upon request to the authors.

*Data availability.* The data of the operational radars was received from the Deutscher Wetterdienst (DWD) archive. The data of the research radar Hohenpeißenberg (MHP) was received from the meteorological research observatory Hohenpeißenberg which is part of the DWD. Recent data of the operational radar network can be accessed on the open data website of the DWD (https://opendata.dwd.de/, last access: 28 January 2025). Radar data not available on the open data website or data of less recent days can only be requested directly from the DWD. Contact is available through kundenservice@dwd.de following DWD customer relations. The data of POLDIRAD is available upon request to the authors. The data of MIRA-35 is available within the ACTRIS Cloudnet data portal (https://cloudnet.fmi.fi/, last access: 28 January 2025).

*Author contributions.* CH developed the methodology, created the necessary software and wrote the original draft of the paper. FE helped with conceptualization of the paper, supervised the methodology development process and proofread the paper. SG supported the conceptualization of the paper and proofread the paper as well.

*Competing interests.* The contact author has declared that none of the authors has any competing interests.

*Acknowledgements.* This research is funded by the Deutsche Forschungsgemeinschaft (DFG, German Research Foundation) in grant number EW 156/1-2: "Investigation of the initiation of convection and the evolution of precipitation using simulations and polarimetric radar observations at C- and Ka-band" (IcePolCKa) within the Priority Program SPP 2115 „Polarimetric Radar Observations meet Atmospheric Modelling (PROM) – Fusion of Radar Polarimetry and Numerical Atmospheric Modelling Towards an Improved Understanding of Cloud and Precipitation Processes".
We thank our colleagues at LMU for the operation and servicing of MIRA-35 as well as for the on-going cooperation.
We thank our colleagues at the meteorological research observatory Hohenpeißenberg for support regarding data of and questions about the operational radar network.

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
