# Peer review of "Augmenting the German weather radar network with vertically pointing cloud radars: implications of resolution and attenuation"

_EGUsphere, 2025_

## Referee Comment (RC1)

Review of egusphere-2025-691

The manuscript's topic is within the journal's scope. The authors introduce a method based on the updated/improved CVP for German weather radar network augmentation. The paper is solidly written, with only a few minor remarks from the reviewer—see the specific comments. The recommendation is acceptance past minor revision.

Specific comments:

Line 79: Rephrase "QVP-radars" to perhaps "radar-centric QVPs", or "radars providing QVPs".

Figure 3: Add the mutual distances between the MIRA and the other radars to the figure for easier visualization (the information is already in the text; e.g., the distance from MIRA to MHP is ~ 57 km).

Figure 4: According to the figure, the POLDIRAD RHI data are not directly over the MIRA-35. What is the offset? The authors should add this fact to the text where appropriate.

Lines 235-310: Is reflectivity averaging performed on the linear or dB scale? Z and ZDR averaging should be on the linear scale.

Line 345: The radar measurements at higher frequencies are affected below the melting layer and in the melting layer, where the melting begins.

Lines 373-374: Rephrase "Germany was located at the front side" to perhaps "Germany was affected by the front side"

Line 400: The authors should add the term "mostly" to "offers gapless" to address the features in Fig. 8 more appropriately.

Line 408: The part of the sentence "slightly bright band…" is confusing. Clarify.

Line 415: The authors should use "fewer" instead of "much less" as it fits the context better.

Lines 418-419: Add commas to the sentence – it is harder to read without them: "Alone from the data of the radar MEM, the ML height can, therefore, only be estimated to lie between 1.2 km and 3 km within the lowest measured radar beam."

Lines 484-485: Rephrase the sentence, it is hard to follow. Perhaps "On this date, more high-ranging clouds upwards of 7 km were present, marking the region where the closer radar ISN has less coverage due to the limited number of measured elevation angles." fits the context better.

Line 495: "DB" should be "dB".

Line 498: There is a subtle increase in Ze (Fig 11b, MHP) before 17:00, between 17:38 and 17:45, and at ~17:54 UTC with respect to 3 km height, indicating ML. However, the ML signal is very difficult to interpret without other measurments.

Line 504: Add "to" between "due the...".

Line 508: Remove one "method" from "profile method method by…".

Lines 553-555: The statement about ice hydrometeor attenuation is questionable. Liquid hydrometeors experience much higher attenuation than ice particles at the C band. Authors may want to investigate the causes of such behavior further. Could the different microphysics along the path explain and account for the difference? Another convective cell along the route, closer to the MEM?

Lines 570-571: Permute the word order in "the with range decreasing sensitivity", it is hard to follow ("the decreasing sensitivity with range" should fit much better).

Line 639: Use "beam-broadening" instead of "-broadening".

Line 641: Use "fewer" instead of "less".

---

## Author Comment (AC1)

The authors thank the reviewer for carefully reading the manuscript and providing numerous suggestions on how to improve the paper. Each comment will be addressed in the following.

**Comment:**
Line 79: Rephrase "QVP-radars" to perhaps "radar-centric QVPs" or "radars providing QVPs".
**Answer:**
Thank you for this suggestion. We have replaced "QVP-radars" with "**radars providing QVPs**"

**Comment:**
Figure 3: Add the mutual distances between the MIRA and the other radars to the figure for easier visualization (the information is already in the text; e.g., the distance from MIRA to MHP is 57 km).
**Answer:**
Thank you for this suggestion. We have added the distances between MIRA-35 and all relevant radars to Figure 3.

**Comment:**
Figure 4: According to the figure, the POLDIRAD RHI data are not directly over the MIRA-35. What is the offset? The authors should add this fact to the text where appropriate.
**Answer:**
Thank you for this observation. While the pointing accuracy of POLDIRAD is monitored, a perfect orientation towards MIRA-35 cannot be guaranteed and comes with a certain pointing error. Considering the distance of 23 km, a pointing error of just 0.25° would lead to roughly 100 m distance between measurement and location of MIRA-35 which is close to the offset visible in this case. We have added the following sentence to the description of the figure to point this out:
*The data of POLDIRAD is not directly above MIRA-35 with a very minor offset which is the result of the pointing error associated with the pointing accuracy of POLDIRAD.*

**Comment:**
Lines 235-310: Is reflectivity averaging performed on the linear or dB scale? Z and ZDR averaging should be on the linear scale.
**Answer:**
All averaging steps are indeed performed in the linear space. We have added the following sentence to make this clear:
*This and all following averaging steps are done in linear space for each variable.*

**Comment:**
Line 345: The radar measurements at higher frequencies are affected below the melting layer and in the melting layer, where the melting begins.
**Answer:**
Thank you for this clarification. We have changed the sentence as follows:
*[...] where the beam remains in or below the melting layer (e.g. at low elevation angles) and therefore encounters mostly liquid hydrometeors [...]*

**Comment:**
Lines 373-374: Rephrase "Germany was located at the front side" to perhaps "Germany was affected by the front side".
**Answer:**
Thank you for this suggestion. We have changed the sentence accordingly.

**Comment:**
Line 400: The authors should add the term "mostly" to "offers gapless" to address the features in Figure 8 more appropriately.
**Answer:**
The word "mostly" indeed helps to capture the features of Figure 8 more appropriately. We have changed the sentence accordingly.

**Comment:**

Line 408: The part of the sentence "slightly bright band..." is confusing. Clarify.

**Answer:**

We understand that the word "slightly" is misleading. The bright band in Figure 9 (b) is clearly distinguishable. We have therefore removed the word "slightly" from the sentence.

**Comment:**

Line 415: The authors should use "fewer" instead of "much less" as it fits the context better.

**Answer:**

Thank you for this correction. We have replaced the phrase accordingly.

**Comment:**

Lines 418-419: Add commas to the sentence - it is harder to read without them: "Alone from the data of the radar MEM, the ML height can, therefore, only be estimated to lie between 1.2 km and 3 km within the lowest measured radar beam.

**Answer:**

Thank you for this suggestion. We have implemented the commas accordingly.

**Comment:**

Lines 484-485: Rephrase the sentence, it is hard to follow. Perhaps "On this date, more high-ranging clouds upwards of 7 km were present, marking the region where the closer radar ISN has less coverage due to the limited number of measured elevation angles." fits the context better.

**Answer:**

Thank you for this suggestion. We have modified the sentence accordingly.

**Comment:**

Line 495: "DB" should be "dB".

**Answer:**

Thank you for this correction!

**Comment:**

Line 498: There is a subtle increase in Ze (Figure 11 (b), MHP) before 17:00, between 17:38 and 17:45 and at   17:54 UTC with respect to 3 km height, indicating ML. However, the ML signal is very difficult to interpret without other measurements.

**Answer:**

Thank for this very detailed observation. We have modified the sentence accordingly:

*Very subtle increases in $Z_e$ shortly before 17:00 UTC, between 17:38 UTC and 17:45 UTC as well as at around 17:54 UTC in roughly 3 km height hint to the position of the ML; they are however, difficult to interpret without the measurements of MIRA-35.*

**Comment:**

Line 504: Add "to" between "due the...".

**Answer:**

Thank you for this correction. We have implemented it accordingly.

**Comment:**

Line 508: Remove one "method" from "profile method method by...".

**Answer:**

Thank you for this correction. We have implemented it accordingly.

**Comment:**

Lines 553-555: The statement about ice hydrometeor attenuation is questionable. Liquid hydrometeors experience much higher attenuation than ice particles at the C band. Authors may want to investigate the causes of such behavior further. Could the different microphysics along the path explain and account for the difference? Another convective cell along the route, closer to the MEM?

**Answer:**

We agree that ice hydrometer attenuation might not be the reason for the observable differences in the BA-CVPs of $\Phi_{dp}$ and the retrieved liquid hydrometeor attenuation using the gate-by-gate approach of Jacobi and Heistermann (2016). Following the suggestion of the reviewer, we examined the beam path more closely for the duration between 08:30 UTC and 10:00 UTC where the BA-CVPs of $\Phi_{dp}$ and the retrieved liquid hydrometeor attenuation differ the most. Several cells with high reflectivity traveling through the beam path are observable which might be able to explain the observed differences. Specifically the influence of the melting layer in convective cells on the radar beams traveling at an angle through the melting layer for a prolonged period might lead to effects that are captured by $\Phi_{dp}$ but not the reflectivity-based approach of Jacobi and Heistermann (2016) valid only for liquid hydrometeor attenuation.

To inlcude these considerations, we have changed the respective text passage to:

*[...] While examining the beam path for MEM during time periods where the extracted $\Phi_{dp}$ BA-CVPs and the calculated liquid hydrometeor attenuation differ significantly (e.g. 08:30 UTC to 10:00 UTC), several cells with strong reflectivity signal traveling through the beam of MEM were observable. The influence of the melting layer of convective cells on the radar beams traveling within the melting layer at an oblique angle over a long distance might lead to effects that are captured by $\Phi_{dp}$ but not the reflectivity-based liquid hydrometer attenuation by Jacobi and Heistermann (2016).*

**Comment:**
Lines 570-571: Permute the word order in "the with range decreasing sensitivity", it is hard to follow ("the decreasing sensitivity with range" should fit much better).
**Answer:**
Thank you for this suggestion. We have modified the sentence accordingly.

**Comment:**
Line 639: Use "beam-broadening" instead of "-broadening".
**Answer:**
Thank you for this suggestion. We added the word "beam" before "-broadening".

**Comment:**
Line 641: Use "fewer" instead of "less".
**Answer:**
Thank you for this suggestion. We have replaced "less" with "fewer".

**References**

Jacobi, S. and Heistermann, M.: Benchmarking attenuation correction procedures for six years of single-polarized C-band weather radar observations in South-West Germany, Geomatics, Natural Hazards and Risk, 7, 1785–1799, https://doi.org/10.1080/19475705.2016.1155080, 2016.

---

## Author Comment (AC2)

The authors thank the reviewer for carefully reading the manuscript and providing numerous suggestions on how to improve the paper. Each comment will be addressed in the following. Please note that the order of the comments has changed slightly as one of the comments has been moved to the end. This was done to allow the image in the respective comment to be displayed at the correct position within the comment it relates to without disturbing the text flow of other comments.

**Comment:**
Line 215-217: I am a bit confused here as the C-band radar resolution is 50 m and the moving average windows size is also 50 m?
**Answer:**
Thank you for this observation. The 50 m are indeed only half of the bin height. The chosen bin height value is 100 m. We have corrected the value given in the manuscript accordingly.

**Comment:**
Line 221: Please elaborate the reasoning as to why all dual-pol variables require Z based weighting? And how is done?
**Answer:**
We understand that using the expression "all" is misleading in this case. The weighting is applied for $Z_{dr}$ and $\Phi_{dp}$ only which we now also point out in the manuscript. It is well known (e.g. Ryzhkov et al. (2016); Ryzhkov and Zrnic (2019)) that noise impacts $Z_{dr}$ or $\Phi_{dp}$ more significantly and is noticeable for even moderate signal-to-noise-ratios. Especially when only a small reflectivity signal is detected, $Z_{dr}$ and $\Phi_{dp}$ can have substantial bias. Since all averaging for the BA-CVP method is done in linear space, the impact of such noise-related biases can be significant. By weighting $Z_{dr}$ and $\Phi_{dp}$ with their reflectivity values during averaging, we ensure that values with high reflectivity and less bias have more weight in the average than values with low reflectivity that might be influenced by noise and therefore carry a bias. Through this averaging procedure, we also try to mimic the behavior of a real radar measuring a volume where the polarimetric return in $Z_{dr}$ and $\Phi_{dp}$ is produced mainly by the hydrometeors with strong reflectivity.

The reflectivity-weighting procedure for the dedicated radars is very similar to the second (beam-aware) averaging step in the BA-CVP method without the weighting factors for area of intersection $w_{b,i}$ and azimuthal averaging $w_{a,i}$. We introduce a normalized weighting factor $w_{z,i}$ for each data point $x_i$ within the current height bin. The weighting factor for each $x_i$ is defined as the reflectivity value of $x_i$ divided by the sum of all reflectivity values contributing within the current height bin.
To make this procedure more transparent, we have added two equations and a short paragraph for explanation in the corresponding section. We have also added a short note in section 3.1.2. where we explain the BA-CVP method referencing this explanation, since the BA-CVP method also utilizes reflectivity weighting as stated previously.

**Comment:**
Line 300: For this paragraph, I am not sure if I understand the BV-CVP extraction for PPI data. The traditional QVP/CVP method consider distance compared with center point of selection. Can you please demonstrate how the new BV-CVP differs from traditional CVP in this point? Most importantly, at higher elevations, the available scans are sparse, do you mean the BV-CVP can fill the gaps? Bus this can only be true in this study's set up when you have more than one radar doing PPI scans nearby right?
**Answer:**
The reviewer correctly points out that the original CVP method considers the contributions of data points within a height bin based on their vertical distance to the center point of the respective height bin. This procedure leads to artificial gaps in the extracted profiles in areas with low data point density even if the original measurement volumes had no gaps in between them (Murphy et al., 2020).
The BA-CVP method counteracts this by weighting each data point by the area of intersection between the beam-broadened range gate and individual height bin instead of just the vertical distance to the height bin center. Data points therefore do not have to physically be within the height bin. It is enough for the measurement volume to intersect the height bin. The data point is then factored into the average according to the size of the intersection reducing the occurrence of artificial gaps.

This beam-aware treatment of each data point furthermore allows contributions of multiple radars in range of the point of interest which as correctly mentioned by the reviewer closes gaps based on measurement strategy. The combination of multiple radars might also be possible within the original CVP method, however, due to the vertical distance weighting, data points with considerable measurement volume mismatch might be weighted equally reducing the achievable resolution.

In order to better demonstrate the main benefits of the BA-CVP method over the original CVP method by Murphy et al. (2020) we have added the following bullet points to the manuscript at the end of section 3.1.2:

- *BA-CVP method considers beam broadening and weighs data points by area of intersection between height bin and radar beam instead of vertical distance to height bin used in the original CVP method*

- *Data points not within a height bin still contribute to the height bin average if measurement volume intersects the height bin, reducing gaps in areas with low data point density*

- *Possibility to combine the data of multiple radars at different distances to the point of interest due to beam-aware treatment of data points, increasing statistical significance and reducing gaps*

- *Gaps in extracted BA-CVPs are real, unmeasured gaps that were not part of any radar measurement volume*

**Comment:**

Line 355-370: More clarification is needed here, how is hydrometeor induced attenuation done? By only Z? Please elaborate on this.

**Answer:**

In this paper we do not correct for any hydrometeor attenuation as the main intention of the paper is to describe the BA-CVP method and show its application in two different case studies. We merely calculate the liquid hydrometeor attenuation using the gate-by-gate approach based on reflectivity by Jacobi and Heistermann (2016) and then compare the calculated liquid hydrometeor attenuation to BA-CVPs of $\Phi_{dp}$ to discuss the usage of the BA-CVPs of $\Phi_{dp}$ as a marker for strong path attenuation. This is done to investigate the possibility to filter out measurement pixels with strong path attenuation that might influence the statistics of the extracted profiles. For more details on the gate-by-gate approach based on reflectivity please refer to Jacobi and Heistermann (2016).

To make this more clear, we have rewritten the corresponding passage in the corresponding section as follows:

*In this paper no hydrometeor attenuation is applied to the measurements. Instead, the calculated liquid hydrometeor attenuation based on the gate-by-gate approach of Jacobi and Heistermann (2016) is compared to the extracted BA-CVPs of $\Phi_{dp}$ and the usage of the $\Phi_{dp}$ BA-CVPs as a qualitative marker to filter out measurement pixels with high attenuation is discussed.*

**Comment:**

Figure 11: For the 2nd case, it seems the higher resolution MHP are overly smoothed to lower resolution as in ISN and MEM, please justify the resolution degradation here.

**Answer:**

As correctly stated by the reviewer, for the second case, the data of the radar MHP was used. MHP and MIRA-35 are 57 km apart. The beams of MHP at the location of MIRA-35 are therefore already broadened to roughly 1 km which leads to natural smoothing of the measurements when compared to the first case where data of POLDIRAD with a distance of only 23 km was used. The processing and moving average performed on the raw RHI data have minimal impact as the bin height is only 100 m. We have added the following sentence to point this out more clearly:

*As MHP is about 2.5 times further away than POLDIRAD, the results of the dedicated measurements are expected to be affected by beam-broadening and therefore appear a bit smoother compared to the first case study.*

**Comment:**

Figure 8: Can you please add the difference between BV-CVP vs. MIRA? Also, please extend this to ZDR, KDP and CC as well. In particular, a vertical profile of mean difference and their standard deviation are needed to quantify the difference.

**Answer:**

Thank your very much for this comment. The idea of adding mean profiles with their standard deviation greatly helps the reader to understand the capabilities of the BA-CVP method in more detail and confirms previous findings.

Since the first case study consists of precipitation in varying intensity with several convective cells traveling over the region of interest, we have added mean profiles of $Z_{\rm e}$, $Z_{\rm dr}$ and the dual-wavelength ratio ($DWR_{\rm C,\,Ka}$) for a more stratiform time period between 08:50 UTC and 09:50 UTC where the $\Phi_{\rm dp}$ of POLDIRAD and the extracted $\Phi_{\rm dp}$ BA-CVPs of the operational radars indicated low path attenuation for best comparability:

[Figure]

Figure 1: Averaged profiles of $Z_{\rm e}$ in (a), $DWR_{\rm C,\,Ka}$ in(b) and $Z_{\rm dr}$ in (c) for a more stratiform time period on 25 May 2019 between 08:50 UTC and 09:50 UTC. The colored area is the standard deviation.

We have also added a paragraph to introduce and discuss the graphic in section 4.1.

Although we agree that plots of $K_{\rm dp}$ and the cross correlation would indeed be interesting from a microphysical perspective, processing and further analysis of these additional radar variables would lie outside of the scope of the paper.

**References**

Jacobi, S. and Heistermann, M.: Benchmarking attenuation correction procedures for six years of single-polarized C-band weather radar observations in South-West Germany, Geomatics, Natural Hazards and Risk, 7, 1785–1799, https://doi.org/10.1080/19475705.2016.1155080, 2016.

Murphy, A. M., Ryzhkov, A., and Zhang, P.: Columnar Vertical Profile (CVP) Methodology for Validating Polarimetric Radar Retrievals in Ice Using In Situ Aircraft Measurements, Journal of Atmospheric and Oceanic Technology, 37, 1623–1642, https://doi.org/10.1175/jtech-d-20-0011.1, 2020.

Ryzhkov, A., Zhang, P., Reeves, H., Kumjian, M., Tschallener, T., Trömel, S., and Simmer, C.: Quasi-

Vertical Profiles—A New Way to Look at Polarimetric Radar Data, Journal of Atmospheric and Oceanic Technology, 33, 551–562, https://doi.org/10.1175/jtech-d-15-0020.1, 2016.

Ryzhkov, A. V. and Zrnic, D. S.: Radar Polarimetry for Weather Observations, Springer International Publishing, ISBN 9783030050931, https://doi.org/10.1007/978-3-030-05093-1, 2019.

---

## Referee Report (RR1)

Review egusphere-2025-691-ATC2

The authors improved the manuscript from the previous version. The recommendation is a minor revision, with suggestions detailed in the specific comments. There is no need for another round of revision after the authors incorporate suggested changes.

Specific comments:

Line 132: Add the following: The discussion is presented in section 5 and, finally, conclusions and final remarks…

Lines 135-137: Rephrase the sentence: "In particular the role of operational radar systems and the use of their data in combination with or as substitute of data measured by dedicated radars is investigated when paired with the data of a vertically pointing radar operating at a different wavelength." with "In particular, the role of operational radar systems and the use of their data in combination with or as a substitute for data measured by dedicated radars is investigated when paired with the data of a vertically pointing radar operating at a different wavelength."

Line 199: Replace "western of" with "west of".

Figure 4 caption: Replace "Shown is the metropolitan region of Munich as a zoom-in of the white rectangle in Fig. 3. The yellow dot marks the position of MIRA-35. Shown are the measurement points of the dedicated (blue colors) and operational radars (red and orange) within their respective segment of roughly 2 km x 2 km centered on MIRA-35 at the lowest elevation angle." with perhaps "The image shows the metropolitan region of Munich as a zoom-in of the white rectangle in Fig. 3. The yellow dot marks the position of MIRA-35. The image also includes the measurement points of the dedicated (blue colors) and operational radars (red and orange) within their respective segment of roughly 2 km x 2 km centered on MIRA-35 at the lowest elevation angle."

Lines 246-247: Rephrase "To still be able to compare the data of a vertically pointing cloud radar to the data collected by operational radars in the form of PPI scans in a time-height manner, BA-CVPs were developed for this study.", perhaps as "BA-CVPs were developed for this study to allow time-height comparison between the data collected by operational radars in the form of PPI scans and the data of a vertically pointing cloud radar."

Line 378: The authors introduce Ze abbreviation without referencing what it is (this is done later, at line 398).

Line 500: The term "smoother" may not be the most adequate here. The resolution is coarser because of the increased distance. Clarify in the text.

Figure 14: The latitude and longitude numbers on all plots are not easily visible. The recommendation is to increase the font size.

---

## Author Response (AR2)

We thank the reviewer for carefully reading the manuscript and for providing numerous suggestions on how to improve the manuscript. Each comment will be addressed in the following.

**Comment:**
Line 132: Add the following: The discussion is presented in section 5 and, finally, conclusions and final remarks...
**Answer:**
Thank you for this suggestion. We have modified the manuscript accordingly.

**Comment:**
Line 135-137: Rephrase the sentence: "In particular the role of operational radar systems and the use of their data in combination with or as substitute of data measured by dedicated radars is investigated when paired with the data of a vertically pointing radar operating at a different wavelength." with "In particular, the role of operational radar systems and the use of their data in combination with or as a substitute for data measured by dedicated radars is investigated when paired with the data of a vertically pointing radar operating at a different wavelength."
**Answer:**
Thank you for this suggestion. We have changed the sentence accordingly.

**Comment:**
Line 199: Replace "western of" with "west of"
**Answer:**
Thank you for this correction. We have implemented it accordingly.

**Comment:**
Figure 4 caption: Replace "Shown is the metropolitan region of Munich as a zoom-in of the white rectangle in Fig. 3. The yellow dot marks the position of MIRA-35. Shown are the measurement points of the dedicated (blue colors) and operational radars (red and orange) within their respective segment of roughly 2 km x 2 km centered on MIRA-35 at the lowest elevation angle." with perhaps "The image shows the metropolitan region of Munich as a zoom-in of the white rectangle in Fig. 3. The yellow dot marks the position of MIRA-35. The image also includes the measurement points of the dedicated (blue colors) and operational radars (red and orange) within their respective segment of roughly 2 km x 2 km centered on MIRA-35 at the lowest elevation angle."
**Answer:**
Thank you for this suggestion. We have modified the caption of Figure 4 accordingly.

**Comment:**
Lines 246-247: Rephrase "To still be able to compare the data of a vertically pointing cloud radar to the data collected by operational radars in the form of PPI scans in a time-height manner, BA-CVPs were developed for this study.", perhaps as "BA-CVPs were developed for this study to allow time-height comparison between the data collected by operational radars in the form of PPI scans and the data of a vertically pointing cloud radar."
**Answer:**
Thank you for this suggestion. We have implemented it accordingly.

**Comment:**
Line 378: The authors introduce Ze abbreviation without referencing what it is (this is done later, at line 398).
**Answer:**
Thank you for this correction. We now fully explain $Z_e$ in line 378 and removed the full explanation from line 398.

**Comment:**
Line 500: The term "smoother" may not be the most adequate here. The resolution is coarser because of the increased distance. Clarify in the text.
**Answer:**

Thank you for this correction. We have removed the term "smoother" and instead mentioned the coarsened resolution due to increased distance.

**Comment:**
Figure 14: The latitude and longitude numbers on all plots are not easily visible. The recommendation is to increase the font size.

**Answer:**
Thank you for this suggestion. We have increased the font size of the axis inscriptions for all plots in Figure 14.

---

## Author Response (AR3)

We thank the editor for again reading the manuscript very carefully and for providing numerous suggestions on how to improve the manuscript. Each comment will be addressed in the following.

**Comment:**
Line 56: I believe this should be "within one scan".
**Answer:**
Thank you for this correction. We have replaced the wrong word "once" with "one".

**Comment:**
"This and the fact that both QVP methods loose spatial information due to the azimuthal averaging that includes the whole 360∘ PPI measurement range, limit the use of QVP methods in combination with other measurement equipment further away from the PPI scanning radar as the extracted QVPs might not fully correspond to the measurements of the off-site equipment." This sentence is too long and very hard to follow. Suggestion: "This combined the fact that both QVP methods loose spatial information due to the 360° azimuthal averaging limit the use of QVP methods in combination with vertically measurement equipment further away from the PPI scanning radar since the extracted QVPs might not adequately represent the measurements of the off-site equipment."
**Answer:**
Thank you for this suggestion. We have implemented it accordingly.

**Comment:**
Legend of Figure 1: "Shown are also all operational cloudnet sites (yellow) and the two dedicated radar systems (blue) utilized for comparison."
**Answer:**
Thank you for this correction. We have added the word "blue" in brackets.

**Comment:**
Line 135: "spatially separated radar systems" (instead of radars).
**Answer:**
Thank you for this correction. We have removed the "s".

**Comment:**
Legend of Figure 2: Add a period at the end.
**Answer:**
Thank you for this correction. We have added a period at the end of the sentence.

**Comment:**
Legend of Figure 3: "Dedicated and operational radars are marked red and blue, respectively." It seems like the colors are reversed in this sentence.
**Answer:**
Thank you for this correction. The colors were indeed reversed. We have changed the description of Figure 3 accordingly.

**Comment:**
Line 310: Suggest change "The method therefore automatically puts more weight in the measured volumes that most closely represent the contents of the height bin."
**Answer:**
Thank you for this correction. We have implemented it accordingly.

**Comment:**
Line 311: Suggest remove "Following". "This procedure allows ..."
**Answer:**
Thank you for this suggestion. We have removed the word "following" from the beginning of the sentence.

**Comment:**

Line 352: Add a period at the end of the sentence.
**Answer:**
Thank you for this correction. We have added a period at the end of the last bullet point.

**Comment:**
Line 422: Add commas: "which, as seen in Fig. 8 (e), offers mostly gapless coverage..."
**Answer:**
Thank you for this correction. We have added the commas accordingly.

**Comment:**
Lines 466-467: "In general, the lowest elevation angles are strongly influenced by clutter and therefore little to no reliable signature of hydrometeors for lower altitudes is measurable." This sentence seems to suggest that the lowest scans are consistently unreliable, which is kind of misleading. It is better to include some additional information saying that this is the case for gates close to the radar but not everywhere.
**Answer:**
Thank you for this suggestion. We have modified the sentence to be less absolute: *The lowest elevation angles are often influenced by clutter and therefore little to no reliable signature of hydrometeors for lower altitudes with range gates close to the radar is measurable.*

**Comment:**
Legend of figure 11: Replace "for a more stratiform time period..." with "for the stratiform time period...".
**Answer:**
Thank you for this correction. We have implemented it accordingly.

**Comment:**
Line 506: Correct the typo "through the measurement region."
**Answer:**
Thank you for this correction. We have added the "h" to the word "trough".

**Comment:**
Line 534: "in roughly 3 km height..." I think "in" is not the correct preposition to use here. Suggest rewriting "around 3 km height...".
**Answer:**
Thank you for this suggestion. We have rephrased the respective part of the sentence: *at heights of about 3 km*

**Comment:**
Line 584: "has to be expected" replace with "was expected."
**Answer:**
Thank you for this correction. We have implemented it accordingly.

**Comment:**
Legend of figure 13: Close brackets after (b). "MHP (b and differential reflectivity in a BA-CVP"
**Answer:**
Thank you for this correction. We have added ")" after "(b".

**Comment:**
Line 684: Replace "even finer details" with "even fine details" (finer suggests a comparison : finer than...)
**Answer:**
Thank you for this correction. We have replace "finer" with "fine".

**Comment:**
Lines 693-694: "over the whole Germany"

**Answer:**

Thank you for this correction. We have added the word "the" before "whole Germany".

**Comment:**

Appendix A: It makes sense to include the corresponding radial resolution and indicate that the different PRF (and radial resolution) correspond to the specific events.

**Answer:**

Thank you for this suggestion. We have added the range resolution for both case studies to the table. Please note that the different PRFs do not correspond to the specific case studies and are not related to the change in range resolution. The operational scan strategy of the DWD includes PRF staggering for the lowest 7 scans. To make this more clear, we have added a short notice to the table description: *Two values for the PRF indicate PRF staggering.*